# FORTUNE: FORMULA-DRIVEN REINFORCEMENT LEARNING FOR SYMBOLIC TABLE REASONING IN LANGUAGE MODELS

## ABSTRACT

Tables are a fundamental structure for organizing and analyzing data, making effective table understanding a critical capability for intelligent systems. While large language models (LMs) demonstrate strong general reasoning abilities, they continue to struggle with accurate numerical or symbolic reasoning over tabular data, especially in complex scenarios. Spreadsheet formulas provide a powerful and expressive medium for representing executable symbolic operations, encoding rich reasoning patterns that remain largely underutilized. In this paper, we propose *Formula Tuning* (Fortune), a reinforcement learning (RL) framework that trains LMs to generate executable spreadsheet formulas for question answering over general tabular data. *Formula Tuning* reduces the reliance on supervised formula annotations by using binary answer correctness as a reward signal, guiding the model to learn formula derivation through reasoning. We provide a theoretical analysis of its advantages and demonstrate its effectiveness through extensive experiments on seven table reasoning benchmarks. *Formula Tuning* substantially enhances LM performance, particularly on multi-step numerical and symbolic reasoning tasks, enabling a 7B model to outperform OpenAI o1 on table understanding. Beyond empirical gains, we present several insights into the role of RL in symbolic table reasoning, highlighting the broader potential of formula-driven RL to advance reasoning capabilities in LMs. Our code can be found at https://anonymous.4open.science/r/Fortune-0597.

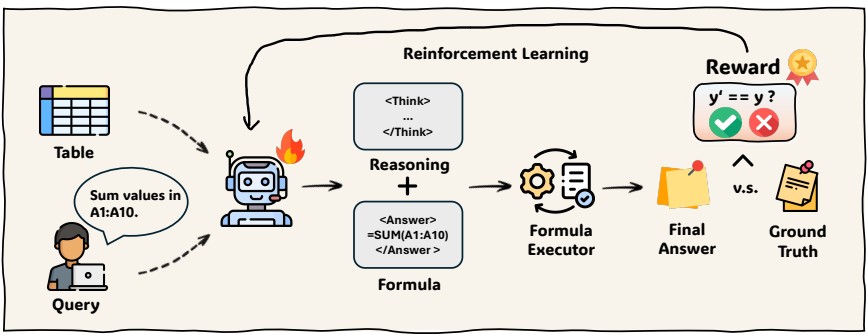

Figure 1: Overview of *Formula Tuning* (FORTUNE).

## 1 INTRODUCTION

Tables are a common and practical data structure in daily life, playing a central role in data collection, representation, and analysis (He et al., 2023; Yi et al., 2025). Recent advances in large language models (LLMs) (Gunasekar et al., 2023; OpenAI, 2024; Touvron et al., 2023) have brought impressive performance across a wide range of natural language processing tasks, including language understanding (Minaee et al., 2024; Zhu et al., 2024) and general reasoning (Plaat et al., 2024). Naturally, LLMs have also been applied to tabular data understanding and reasoning (Fang et al., 2024; Zhang et al., 2024b; Cao & Liu, 2025).

However, reasoning over tabular data remains a key challenge for language models (LMs) (Cao & Liu, 2025), primarily because of the numerical nature and intricate structure of tables, which pose significant obstacles to understanding both content and layout. Besides, some findings (Yan et al., 2025) suggest current LLMs rely on pattern memorization over genuine rule learning, often leading to incorrect mathematical computations during traditional chain-of-thought reasoning, also referred to as textual reasoning. Some approaches introduce symbolic methods (Chen et al., 2023; Gao et al., 2023), where models first generate symbolic representations (such as programs) and then execute them to obtain results. While this improves arithmetic accuracy, such methods often struggle to generalize due to limited symbolic reasoning or program generation capabilities. Instead of truly understanding the context and generating problem-solving programs, LMs often fall back on memorized code snippets from pretraining (Yang et al., 2024). Since not all complex symbolic patterns can be memorized, and given that high-quality code supervision is scarce (Bi et al., 2023), more effective strategies for symbolic table reasoning are needed.

Recent progress in reinforcement learning (RL) for LMs shows promise for overcoming such limitations. For example, DeepSeek-R1 (DeepSeek-AI et al., 2025) improves mathematical reasoning in LLMs via rule-based RL rewards, without relying on step-by-step annotations. DeepRetrieval (Jiang et al., 2025) uses retrieval metrics as RL rewards to train models to reason over queries that maximize real-world retrieval performance across search engines and databases. DeepCoder (Luo et al., 2025) also demonstrates the effectiveness of RL for code reasoning and generation. These works collectively demonstrate the promise of RL in enabling LMs to perform robust symbolic reasoning without explicit intermediate supervision.

When it comes to symbolic table reasoning, **spreadsheet formula** (Microsoft Corporation, 2025) is a powerful and versatile tool. In real-world scenarios, tabular data is often stored in spreadsheet formats (e.g., Microsoft Excel, Google Sheets) (Dong et al., 2024), where each cell holds individual data values. The spreadsheet formulas embedded in these files act as lightweight, program-like constructs that enable users to compute, transform, and reason over data. Compared to structured interfaces such as SQL or Python/Pandas, which are typically restricted to relational or flat table schemas, spreadsheet formulas offer greater flexibility, as they can be applied to arbitrary two-dimensional tables without structural constraints (Wang et al., 2025). Moreover, spreadsheet formulas are Turing complete (Smalley, 2023), making them particularly well-suited as a medium for symbolic table reasoning.

We envision that by training LMs to understand and generate spreadsheet formulas, they can acquire more robust and generalizable symbolic reasoning capabilities over tabular data. However, current LLMs still struggle to produce accurate and reliable spreadsheet formulas, as highlighted by recent evaluations (Thorne, 2023). At the same time, existing publicly available spreadsheet datasets tend to include relatively simple formulas (Cheng et al., 2021), rely on heuristic conversions from SQL (Zhao et al., 2024a), or synthesize formulas using LLMs in constrained question-answering settings (Wang et al., 2025). These approaches fall short of capturing the complexity necessary for diverse symbolic reasoning tasks and real-world downstream applications. This limitation poses a significant barrier to effectively leveraging spreadsheet formulas for training LMs in symbolic table reasoning.

To address these challenges, we propose *Formula Tuning* (**Fortune**), a RL framework designed to teach LMs to perform symbolic reasoning over general tabular data through spreadsheet formula. Specifically, our framework leverages answer correctness of formula execution results as a reward signal to guide the LMs in deriving formulas through reasoning (Figure 1). This approach reduces reliance on supervised formula annotations and enables LMs to generate executable formulas that answer questions over tables with improved accuracy. Extensive experiments validate the effectiveness of *Formula Tuning*, demonstrating that RL is more effective than supervised fine-tuning (SFT) in enhancing the symbolic reasoning capabilities of LMs. We also find that initializing RL with SFT as a cold start (DeepSeek-AI et al., 2025) provides a stronger foundation and raises the upper bound of RL performance, with SFT serving as a form of knowledge injection. Notably, this enables a 7B model to outperform OpenAI o1 in overall performance *(68.48% vs. 66.90%)*. Furthermore, we train both textual and symbolic reasoning components using RL in **Fortune++**. By jointly leveraging both components during inference, our method achieves strong performance across multiple benchmarks *(e.g., 82.54% on WikiTQ, 95.06% on TabFact, 87.24% on HiTab, 80.47% on FinQA, and 93.20% on AIT-QA)*.

In summary, this paper makes the following key contributions:

- We propose *Formula Tuning* (**Fortune**), a reinforcement learning framework that enhances symbolic reasoning for table understanding by training language models to generate executable spreadsheet formulas.
- We provide a theoretical analysis and discussion comparing textual versus symbolic reasoning in table understanding, as well as supervised fine-tuning versus reinforcement learning in symbolic table reasoning.
- We conduct extensive experiments on seven table understanding benchmarks, demonstrating the effectiveness of *Formula Tuning*, and perform comprehensive analyses to provide deeper insights.

## 2 RELATED WORK

**Table Understanding and Reasoning.** Many studies have explored fine-tuning language models (LMs) to improve their ability to understand and reason over tabular data. Building on the masked language modeling introduced by BERT (Devlin et al., 2019), models such as TaPas (Herzig et al., 2020), PaSTA (Gu et al., 2022), and TUTA (Wang et al., 2021) propose specialized pretraining strategies tailored for tables. TAPEX (Liu et al., 2022) pretrains an encoder-decoder model as a neural SQL executor to better capture the semantics of table operations. Recent efforts, including TableLLaMA (Zhang et al., 2024a) and TableGPT (Zha et al., 2023), build upon large decoder-only language models pretrained for general-purpose table understanding across a wide range of downstream tasks.

Other studies focus on enabling LMs to better perform table-related tasks without fine-tuning. For example, Dater (Ye et al., 2023) proposes strategies for dynamically constructing sub-tables, modifying the input context to enhance comprehension. Chain-of-Table (Wang et al., 2024) models table reasoning as a sequence of transformations using predefined operations, gradually generating sub-tables to support complex multi-step inference. TableMaster (Cao & Liu, 2025) introduces a general framework for table understanding and underscores the importance of symbolic reasoning in handling complex scenarios. Given the structured and often numerical nature of tabular data, program-of-thought prompting (Chen et al., 2023) and other symbolic approaches (Cheng et al., 2023; Nahid & Rafiei, 2024; Mao et al., 2024) have demonstrated strong effectiveness for table reasoning.

**Formula Learning.** A growing body of research has explored the potential of spreadsheet formulas as a powerful means to enhance table understanding. NL2Formula (Zhao et al., 2024a) constructs a formula generation dataset by converting text-to-SQL tasks into spreadsheet formulas, enabling position-aware reasoning from natural language queries. ForTap (Cheng et al., 2022) leverages spreadsheet formulas as pretraining signals to enhance numerical reasoning. Auto-Formula (Chen et al., 2024) applies contrastive learning to transfer formulas from similar spreadsheets for formula recommendation. SpreadsheetCoder (Chen et al., 2021a) formulates formula prediction as a program synthesis task, leveraging both headers and surrounding cell values for context. FLAME (Joshi et al., 2023) trains a small domain-specific model tailored for formula repair and completion. TabAF (Wang et al., 2025) jointly generates answers and formulas for table question answering, but relies on supervised fine-tuning over datasets generated by LLMs.

**Reinforcement Learning for Language Models.** Reinforcement Learning (RL) (Kaelbling et al., 1996) is a machine learning paradigm that trains agents to make decisions through interaction with an environment, with the goal of maximizing cumulative rewards. In the era of large language models (LLMs), RL has gained significant traction as an effective framework for aligning models with human preferences. A prominent example is Reinforcement Learning from Human Feedback (RLHF) (Christiano et al., 2017; Stiennon et al., 2020; Ouyang et al., 2022), which leverages the Proximal Policy Optimization (PPO) algorithm (Schulman et al., 2017) and human preference data to train a reward model that guides the fine-tuning of LLMs. Building on RLHF, more recent algorithms such as GRPO (Shao et al., 2024) and REINFORCE++ (Hu, 2025) aim to enhance reward modeling and mitigate issues like biased optimization (Xu et al., 2024).

**Reasoning with Language Models.** It has been observed that sufficiently large language models (LMs) can demonstrate emergent reasoning capabilities (Wei et al., 2022; Suzgun et al., 2022). Chain-of-thought prompting (Wei et al., 2023) is one technique used to elicit step-by-step reasoning,

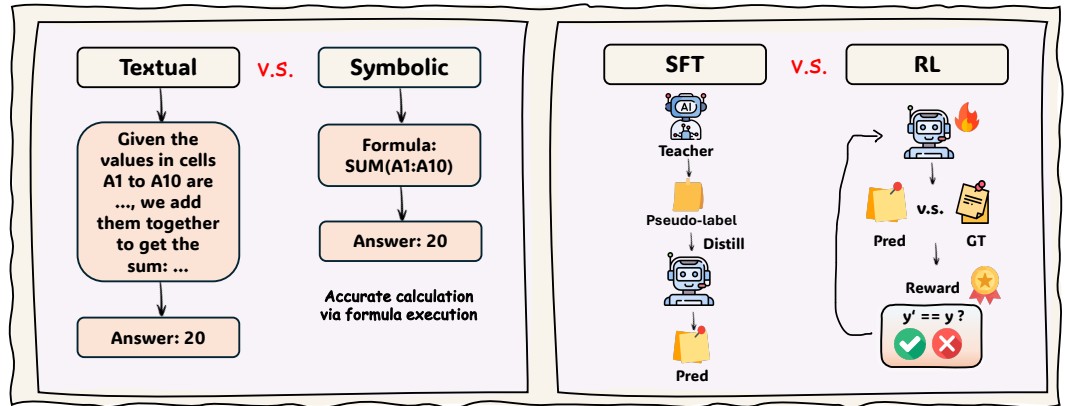

Figure 2: A simplified illustration contrasting Textual versus Symbolic Reasoning in Table Understanding, and Supervised Fine-Tuning (SFT) versus Reinforcement Learning (RL) in Symbolic Table Reasoning.

significantly improving performance on complex tasks. Further advances include self-consistency (Wang et al., 2023) and structuring the reasoning process in forms like trees (Yao et al., 2023) or graphs (Besta et al., 2024; Cao, 2024) are also useful for more complex reasoning tasks. RL has also been used to directly improve reasoning skills during training (Lightman et al., 2023; Uesato et al., 2022). Notably, DeepSeek-R1 (DeepSeek-AI et al., 2025) demonstrates that large-scale RL can substantially boost the reasoning abilities of LMs. In terms of application, DeepRetrieval (Jiang et al., 2025) applies RL to teach models how to reason about interacting with search engines for information retrieval, while DeepCoder (Luo et al., 2025) uses RL for code reasoning and generation tasks. Rec-R1 (Lin et al., 2025) also bridges LLMs and recommendation systems through RL.

## 3 METHODOLOGY

In this section, we present a theoretical analysis and discussion comparing textual versus symbolic reasoning in table understanding, as well as supervised fine-tuning (SFT) versus reinforcement learning (RL) in symbolic table reasoning (Figure 2). We then introduce our proposed training framework, *Formula Tuning*. All notations are list at Appendix L.

### 3.1 TASK FORMULATION

**Table Understanding with Language Models.** We consider a language model (LM) as a conditional generation policy $\pi_\theta(a \mid s)$, where $\theta$ denotes its parameters. The input $s \in \mathcal{S}$ comprises a table $\mathbb{T}$ and a natural-language query $q$, i.e., $s = (\mathbb{T}, q)$. The table $\mathbb{T}$ is a two-dimensional grid of cells:

$$\mathbb{T}_{m \times n} = \begin{bmatrix} C_{1,1} & C_{1,2} & \cdots \\ C_{2,1} & C_{i,j} & \cdots \\ \vdots & \vdots & \ddots \end{bmatrix},$$  (1)

where each $C_{i,j}$ may contain a data value, structural information (e.g., a top header or a left header), or be empty. In practice, we linearize $\mathbb{T}$ into a text sequence before feeding it to the LM.

The LM then generates an output $a \in \mathcal{A}$, which can be either a final textual answer or a spreadsheet formula $f$ that produces the answer upon execution. Our goal is to optimize the parameters $\theta$ that maximize the expected table-understanding performance, measured by a reward function $r(a \mid s)$. Formally,

$$\max_\theta \ \mathbb{E}_{s \sim p(s),\, a \sim \pi_\theta(\cdot \mid s)} \big[ r(a \mid s) \big],$$  (2)

where $p(s)$ denotes the empirical distribution over table–query pairs and $r(a \mid s)$ evaluates the correctness of the final answer from the LM given the input.

## 3.2 TEXTUAL VS. SYMBOLIC REASONING IN TABLE UNDERSTANDING

**Definition 1** (Textual and Symbolic Policies). Given an input $s = (\mathbb{T}, q)$, we consider two types of reasoning strategies:

1. **Textual policy** $\pi_\theta^{\text{txt}}$: The language model generates a chain of thought and directly produces a textual answer $a \in \mathcal{A}_{\text{txt}}$.

2. **Symbolic policy** $\pi_\theta^{\text{sym}}$: The language model generates a chain of thought followed by a spreadsheet formula $f \in \mathcal{F}$; the final answer is obtained by executing the formula deterministically: $a = \text{exec}(f, \mathbb{T})$.

**Theorem 1** (Symbolic Reasoning Superiority). *Under mild assumptions, the expected reward achieved by symbolic reasoning is greater than or equal to that of textual reasoning for any input s:*

$$\mathbb{E}_{a \sim \pi_\theta^{\text{sym}}}[r(a \mid s)] \geq \mathbb{E}_{a \sim \pi_\theta^{\text{txt}}}[r(a \mid s)]. \tag{3}$$

The assumptions and the proof of Theorem 1 are provided in Appendix C.1.

*Remark* 1 (Symbolic Reasoning Potential Benefit). Maximizing the expected reward in Eq. equation 2 therefore tends to favor the symbolic policy $\pi_\theta^{\text{sym}}$ over the textual policy $\pi_\theta^{\text{txt}}$. Symbolic reasoning is particularly advantageous for complex tables and questions requiring multi-step computation or precise numerical manipulation, since correctness is determined by the execution result rather than the exact reasoning trace. As a result, it often achieves higher accuracy than purely textual reasoning.

## 3.3 SFT VS. RL IN SYMBOLIC TABLE REASONING

**Theorem 2** (RL Superiority). *Under mild assumptions, and assuming the reward function $r(a \mid s)$ is reasonably aligned with task success (e.g., exact match), reinforcement learning (RL) can in principle attain higher expected reward than supervised fine-tuning (SFT):*

$$\mathbb{E}_{s \sim p, \, a \sim \pi_\theta^{\text{RL}}}[r(a \mid s)] \geq \mathbb{E}_{s \sim p, \, a \sim \pi_{\theta^\star}^{\text{SFT}}}[r(a \mid s)]. \tag{4}$$

The assumptions and the proof of Theorem 2 are provided in Appendix C.2.

*Remark* 2 (RL Objective and Potential Benefit). Unlike SFT, which is constrained to imitating the teacher policy $\pi_g$, reinforcement learning (RL) directly seeks to maximize the expected task reward:

$$\max_\theta \mathbb{E}_{s \sim p(s), \, a \sim \pi_\theta(\cdot|s)}[r(a \mid s)]. \tag{5}$$

This objective may allow the model to assign probability mass to high-reward actions that lie outside the support of $\pi_g$—for example, alternative formulas that yield the correct answer but differ syntactically or structurally from those observed during supervised training.

In symbolic table reasoning, such flexibility can be particularly helpful: since many distinct formulas can yield the same correct result, RL may leverage this many-to-one mapping by exploring diverse yet semantically valid expressions. Consequently, RL has the potential to surpass the SFT reward bound, under mild assumptions on reward alignment and exploration quality.

## 3.4 FORMULA TUNING

**Definition.** *Formula Tuning* is a reinforcement learning (RL) framework that defines spreadsheet formulas as an explicit symbolic reasoning space for table understanding. Specifically, we fine-tune a pretrained LM $\pi_\theta$ to generate formulas $f \in \mathcal{F}$, which are executed by a deterministic spreadsheet engine $\text{exec}(f, \mathbb{T})$. The resulting answer $a = \text{exec}(f, \mathbb{T})$ is compared against the ground-truth answer $a^\star(s)$, and the model receives a reward:

$$r(a \mid s) = \begin{cases} 1, & \text{if } a = a^\star(s), \\ 0.2, & \text{if } a \neq a^\star(s) \text{ and } f \text{ is executable,} \\ 0, & \text{if } f \text{ is not executable.} \end{cases} \tag{6}$$

This reward function encourages the model to explore valid, executable formulas—even if initially incorrect—while assigning full credit only when the answer exactly matches the ground truth.

**Objective.** Formula Tuning maximizes the expected reward using RL algorithms, such as proximal policy optimization (PPO) (Schulman et al., 2017), with the action space constrained to spreadsheet formulas:

$$\max_{\theta} \; \mathbb{E}_{s \sim p(s), \, f \sim \pi_\theta(\cdot|s)} \left[ r \left( \exp(f, \mathbb{T}) \mid s \right) \right]. \tag{7}$$

**Training Workflow.**

1. *Decoding:* The LM generates a chain of thought and samples candidate formulas $f_1, f_2, \ldots$ from its current policy $\pi_\theta$.
2. *Execution:* Each formula $f_k$ is executed to produce the corresponding answer $a_k$.
3. *Rewarding:* The environment returns a scalar reward $r_k = r(a_k \mid s)$ based on the correctness and executability of the result.
4. *Policy Update:* The LM parameters $\theta$ are updated using a RL algorithm (e.g., PPO) based on the observed tuple $(s, f_k, r_k)$.

This framework enables the model to perform symbolic reasoning over general tables with higher accuracy, with its advantages analyzed earlier. For additional discussion of the methodology, please refer to Appendix D.

## 4 EXPERIMENTS

### 4.1 SETTINGS

We conduct experiments on seven diverse table understanding benchmarks, including WikiTQ (Pasupat & Liang, 2015), TabFact (Chen et al., 2020), FinQA (Chen et al., 2021b), HiTab (Cheng et al., 2021), MultiHiertt (Zhao et al., 2022), AIT-QA (Katsis et al., 2021), and TableBench (Wu et al., 2025). These datasets differ in domain sources, table structure types, and question complexity, collectively covering the full spectrum of table understanding tasks. For training, we merge the first five datasets into a single training corpus and train the model jointly on this combined set, then evaluate it separately on each dataset. Among these, AIT-QA and TableBench are treated as out-of-distribution (OOD) evaluation sets, while the rest are considered in-distribution (ID). Our experiments cover a range of models, including *GPT-4o-mini*, *GPT-4o*, *O1*, *Llama-3.1$_{8B}$*, and *Qwen2.5-Coder$_{7B}$*. Following prior work (Pasupat & Liang, 2015; Cheng et al., 2021), we use exact match accuracy as our primary evaluation metric. The prompts used in our experiments are provided in Appendix K. Detailed settings are provided in Appendix E.

### 4.2 PERFORMANCE OF FORMULA LEARNING UNDER SFT, RL, AND COLD-START RL

Table 1 presents the performance of different formula learning methods under supervised fine-tuning (*SFT*), reinforcement learning (*RL*), RL with a cold-start strategy (*RL w/ CS*), and direct zero-shot inference without any training. This experiment is primarily designed to validate the theoretical analysis discussed earlier and to discuss how its behavior in practical scenarios aligns with our theoretical analysis. Several key analyses and insights are summarized below:

**Large zero shot performance gap between textual and symbolic reasoning.** Closed-source models such as *GPT-4o* achieve an overall accuracy of 66.51% using purely textual reasoning, yet their accuracy on formula based tasks drops significantly to 58.50%. The gap is even more pronounced in 7B and 8B open-source models. Notably, *Qwen2.5-Coder$_{7B}$*, which benefits from additional code pretraining, shows a modest ability to generate formulas. This reinforces the observation that vanilla pretraining leaves models largely unaware of spreadsheet syntax and semantics. These findings highlight the critical need for *formula tuning* to bridge the symbolic reasoning gap.

**SFT rapidly narrows the textual and symbolic reasoning gap.** After SFT, open-source models gain 17–20 accuracy points (e.g. *Llama-3$_{8B}$* rises from 41.47% to 58.71%), and the residual gap

Table 1: Performance under Zero-shot, SFT, and RL settings across models and datasets. Values in the table indicate accuracy (%). *Text* and *Formula* refer to textual and symbolic reasoning methods, respectively. *w/ CS* denotes cold-start RL initialized from SFT. For open-source models, the best performance in each column is highlighted in dark blue, and the second-best in light blue.

| Base Model | Method | In-Distribution | | | | | Out-of-Distribution | | Overall |
|---|---|---|---|---|---|---|---|---|---|
| | | WikiTQ | TabFact | FinQA | HiTab | MultiHiertt | AIT-QA | TableBench | |
| *Zero-Shot* | | | | | | | | | |
| GPT-4o-mini | Text | 67.36 | 88.44 | 59.20 | 57.89 | 22.41 | 77.67 | 36.79 | 58.54 |
| | Formula | 49.16 | 74.90 | 43.07 | 48.17 | 28.26 | 75.53 | 34.65 | 50.53 |
| GPT-4o | Text | 78.57 | 94.52 | 63.12 | 69.26 | 36.11 | 81.36 | 42.66 | 66.51 |
| | Formula | 53.36 | 79.94 | 48.65 | 65.40 | 38.41 | 85.83 | 37.92 | 58.50 |
| O1 | Text | 77.90 | 95.50 | 55.71 | 74.31 | 38.31 | 81.55 | 45.03 | 66.90 |
| | Formula | 70.40 | 91.11 | 42.81 | 75.00 | 48.95 | 88.93 | 44.02 | 65.89 |
| Llama-3.1$_{8B}$ | Text | 50.46 | 67.84 | 42.37 | 29.92 | 18.77 | 59.61 | 21.29 | 41.47 |
| | Formula | 12.58 | 15.77 | 22.67 | 6.57 | 4.23 | 20.00 | 10.31 | 13.16 |
| Qwen2.5-Coder$_{7B}$ | Text | 55.53 | 78.26 | 55.54 | 50.38 | 26.11 | 73.60 | 24.12 | 51.93 |
| | Formula | 38.96 | 52.52 | 34.44 | 32.60 | 15.90 | 52.43 | 26.05 | 36.13 |
| *Supervised Fine-Tuning (SFT)* | | | | | | | | | |
| Llama-3.1$_{8B}$ | Text | 66.15 | 82.95 | 50.04 | 70.27 | 39.98 | 78.45 | 23.10 | 58.71 |
| | Formula | 59.62 | 72.48 | 58.50 | 72.54 | 44.00 | 74.95 | 31.48 | 59.08 |
| Qwen2.5-Coder$_{7B}$ | Text | 65.98 | 81.13 | 59.46 | 72.35 | 43.93 | 81.55 | 24.01 | 61.20 |
| | Formula | 63.46 | 78.16 | 58.33 | 71.83 | 42.68 | 76.12 | 35.56 | 60.88 |
| *Reinforcement Learning (RL)* | | | | | | | | | |
| Llama-3.1$_{8B}$ | Text | 64.37 | 82.16 | 62.60 | 68.81 | 31.99 | 82.91 | 28.08 | 60.13 |
| | Text w/ CS | 71.56 | 87.01 | 56.84 | 77.64 | 49.34 | 85.66 | 28.16 | 65.17 |
| | Formula | 57.64 | 80.09 | 60.85 | 67.93 | 29.40 | 80.78 | 30.69 | 58.20 |
| | Formula w/ CS | 70.49 | 83.04 | 71.99 | 79.29 | 54.55 | 81.29 | 36.64 | 68.18 |
| Qwen2.5-Coder$_{7B}$ | Text | 66.95 | 85.43 | 64.34 | 74.24 | 35.55 | 85.28 | 27.86 | 62.80 |
| | Text w/ CS | 71.31 | 86.07 | 64.77 | 77.42 | 54.25 | 85.43 | 25.94 | 66.46 |
| | Formula | 67.80 | 84.19 | 62.16 | 71.19 | 41.72 | 81.17 | 35.45 | 63.38 |
| | Formula w/ CS | 70.90 | 86.18 | 69.21 | 77.89 | 56.78 | 79.14 | 39.25 | 68.48 |

between textual and symbolic reasoning shrinks to only 1–2 points. SFT thus injects essential task knowledge and brings symbolic reasoning almost on par with textual reasoning.

**RL yields further gains, especially for formula and OOD.** Starting from scratch, RL substantially improves formula accuracy for both backbones, pushing overall performance above 63% for *Qwen2.5-Coder$_{7B}$* in particular. The improvements for text-only reasoning are comparatively smaller, suggesting that RL primarily enhances the model's ability to generate correct formulas rather than improve surface-level responses. Furthermore, the out-of-distribution results on AIT-QA and TableBench show substantial gains over SFT, demonstrating the generalization benefits of RL.

**Cold-start RL is essential for open-source models.** Initializing RL from an SFT model (the *w/ CS* rows) instead of from scratch delivers an additional 4–10 point lift. The *Formula w/ CS* setting consistently achieves the best open-source numbers: 68.18% for *Llama-3$_{8B}$* and 68.48% for *Qwen2.5-Coder$_{7B}$*. These results suggest that SFT provides a strong knowledge foundation, while RL drives performance closer to its upper bound.

**Symbolic reasoning shows markedly higher robustness on complex benchmarks.** On the challenging TableBench OOD set, *Formula w/ CS* lifts *Llama-3$_{8B}$* from a 10.31% zero-shot score to 36.64%, surpassing *Text w/ CS* by 8 points. This demonstrates that the advantages of symbolic reasoning are most evident on datasets that demand more complex and multi-step computations.

**Textual reasoning remains preferable for simple look-up QA.** On TabFact, *Text w/ CS* outperforms formula-based reasoning (87.01% vs. 83.04% for *Llama-3$_{8B}$*), suggesting that composing an explicit formula is not always beneficial when a direct textual response suffices. A similar trend is observed on AIT-QA, where *Text w/ CS* again achieves higher accuracy (85.66% vs. 81.29%). These results indicate that textual reasoning is more effective in scenarios where answers can be directly extracted from the table without the need for symbolic composition.

**Formula-tuned small models could surpass closed-source LMs.** After SFT + RL, small open-source models achieve strong overall accuracies (68.18% and 68.48%). The performance surpasses

Table 2: Performance comparison of different methods. Values in the table indicate accuracy (%). Values marked with * indicate out-of-distribution results. '-' indicates results not reported in the related paper. For fine-tuning-based methods, the best performance in each column is highlighted in dark blue, and the second-best in light blue.

| Method | Backbone | WikiTQ | TabFact | HiTab | FinQA | AIT-QA |
|---|---|---|---|---|---|---|
| *Prompting-Based Methods* | | | | | | |
| Binder (Cheng et al., 2023) | CodeX | 64.60 | 85.10 | - | - | - |
| Dater (Ye et al., 2023) | CodeX | 65.90 | 85.60 | - | - | - |
| API-Assisted (Cao & Liu, 2025) | CodeX | 42.40 | - | 69.30 | - | - |
| ReAcTable (Zhang et al., 2023) | CodeX | 68.00 | 86.10 | - | - | - |
| Chain-of-Table (Wang et al., 2024) | PaLM 2 | 67.31 | 86.61 | - | - | - |
| Norm-DP&Agent (Liu et al., 2023) | GPT-3.5 | 73.65 | 88.50 | - | - | - |
| TIDE DP&Agent (Yang et al., 2025) | GPT-3.5 | 75.00 | 89.82 | - | - | - |
| TableMaster (Cao & Liu, 2025) | GPT-4o-mini | 78.13 | 90.12 | - | 66.40 | - |
| E5 (Zhang et al., 2024c) | GPT-4 | - | - | 85.08 | - | - |
| SS-CoT (Zhao et al., 2024b) | Llama-3.1$_{70B}$ | 76.80 | - | 79.10 | - | - |
| *Finetuning-Based Methods* | | | | | | |
| FORTAP (Cheng et al., 2022) | BERT+LSTM | - | - | 47.00 | - | - |
| TAPEX-Large (Liu et al., 2022) | BART$_{Large}$ | 59.10 | 84.20 | 45.60 | - | - |
| OmniTab (Jiang et al., 2022) | BART$_{Large}$ | 62.80 | - | - | - | - |
| TableLlama (Zhang et al., 2024a) | Llama-2$_{7B}$ | 32.14* | 82.55 | 60.48 | 2.27* | 26.99* |
| TableLLM (Zhang et al., 2025) | Qwen2$_{7B}$ | 53.59 | 69.81 | 43.88 | 8.63* | 64.85 |
| TableGPT2 (Su et al., 2024) | Qwen2.5$_{7B}$ | 61.42 | 77.80 | 70.27 | 40.28* | 12.43* |
| TabAF (Wang et al., 2025) | Qwen2.5-Coder$_{7B}$ | 74.72 | 83.99 | 78.41 | 45.07* | 62.33* |
| **Fortune (Ours)** | Qwen2.5-Coder$_{7B}$ | 67.05 | 85.08 | 69.74 | 62.16 | 80.39* |
| **Fortune++ (Ours)** | Qwen2.5-Coder$_{7B}$ | 82.54 | 95.06 | 87.24 | 80.47 | 93.20* |

GPT series models and even outperforms the large reasoning model *O1* with 66.9% accuracy. These results highlight the power of *formula tuning* in democratizing high-quality table reasoning.

### 4.3 PERFORMANCE OF FORTUNE AND FORTUNE++ COMPARED TO OTHER METHODS

Table 2 presents the performance of Fortune and Fortune++ compared to several strong baselines, as detailed in Appendix E. Fortune is derived from the best overall performance achieved through cold-start RL in formula-based symbolic reasoning. Following prior work (Liu et al., 2023; Yang et al., 2025; Wang et al., 2025), we adopt the self-consistency strategy (Wang et al., 2023) to enhance table understanding performance. This strategy involves generating multiple candidate formulas and selecting the final answer based on majority voting. It is a widely adopted and effective approach for improving accuracy. **Fortune** follows previous work by generating *10 symbolic reasoning outputs*. To further leverage the complementary strengths of textual and symbolic reasoning, we introduce **Fortune++**, which produces *a balanced mix of 5 textual and 5 symbolic outputs*. Neither method relies on a cold-start strategy.

**Fortune++ delivers consistently strong performance across benchmarks.** Fortune++ surpasses all finetuning-based methods across the reported datasets. Specifically, it achieves 80.47% on FinQA, demonstrating strong complex mathematical reasoning ability. On AIT-QA, Fortune++ brings an improvement of 30.9 points, highlighting the out-of-distribution robustness enabled by RL. These results also show that smaller open-source models can outperform larger closed-source models. Despite using only a 7B-parameter *Qwen* backbone, Fortune++ consistently outperforms nearly all prompting-based methods, including those powered by GPT-4o. The original Fortune, which relies solely on formula-based reasoning, also achieves competitive performance across benchmarks.

**RL surpasses SFT.** TabAF (Wang et al., 2025) is a strong baseline that uses SFT for symbolic reasoning with formula and similarly adopts a hybrid self-consistency strategy with 5 textual and 5 formula-based outputs. Nevertheless, Fortune++ significantly outperforms TabAF, demonstrating that RL offers clear advantages over SFT-only models distilled from stronger teacher models.

**Additional results and further analysis.** Fortune achieves 40.85% on MultiHiertt and 35.22% on TableBench, while Fortune++ achieves 51.73% and 44.96%, respectively. An **ablation study** and upper-bound performance analysis of Fortune and Fortune++ are presented in Appendix F.

Table 3: Performance comparison of different symbolic reasoning methods (SQL, Python, and Formula) under Zero-shot and RL settings. Values in the table indicate accuracy (%). The best performance in each column is highlighted in dark blue.

| Method | WikiTQ | TabFact | FinQA | TableBench | Overall |
|---|---|---|---|---|---|
| *Zero-Shot* | | | | | |
| SQL | 17.07 | 21.15 | 1.51 | 7.64 | 11.84 |
| Python | 30.96 | **60.39** | **34.87** | 16.33 | 35.64 |
| **Formula** | **38.96** | 52.52 | 34.44 | **26.05** | **37.99** |
| *Reinforcement Learning (RL)* | | | | | |
| SQL | 67.58 | 83.94 | 38.79 | 32.09 | 55.60 |
| Python | 70.46 | 84.42 | 65.85 | 35.26 | 64.00 |
| **Formula** | **70.67** | **84.50** | **65.89** | **35.84** | **64.23** |

A **statistical analysis** of the generated formulas is provided in Appendix H, and qualitative **case studies** are discussed in Appendix I. An impact analysis of the **reasoning process** during formula tuning appears in Appendix G.

## 5 COMPARATIVE ANALYSIS OF FORMULA TUNING AND OTHER SYMBOLIC TABLE REASONING METHODS

In this section, we analyze the use of formulas as a symbolic reasoning tool, and compare them with SQL and Python. Table 3 compare three symbolic reasoning paradigms (SQL queries, Python snippets, and spreadsheet formulas) under both zero-shot and reinforcement learning (RL) settings. To ensure a fair comparison, we train and evaluate these symbolic tools only on datasets with flat, relational tables: WikiTQ, TabFact, and FinQA for training, and all three plus TableBench for evaluation. This restriction is necessary because SQL operates solely on flat tables, while Python/Pandas are also usually worked for flat tabular structures. Several patterns emerge from the comparison.

**Spreadsheet formulas offer the strongest zero-shot symbolic reasoning.** Without any task-specific training, formulas achieve the highest out-of-the-box performance, with an overall accuracy of 37.99%. They slightly outperform Python (35.64%) and significantly surpass SQL (11.84%). The gap is especially pronounced on datasets like WikiTQ and FinQA, suggesting that pre-trained language models possess some intuitive understanding of spreadsheet-style operations, but struggle to generate valid and executable SQL or Python code without further adaptation. On TableBench, which features complex table QA questions, Python code falls short. Models without fine-tuning often fail to generate long and sufficiently accurate code to solve challenging problems. In contrast, spreadsheet formulas are shorter, easier to generate, and more robust in zero-shot settings, making them better suited for this type of reasoning.

**Spreadsheet formulas remain the most effective symbolic tool after RL.** All three symbolic tools improve significantly with RL training, with SQL and Python gaining 43.8 and 28.4 percentage points, respectively. This underscores the value of policy-gradient optimization for learning execution-constrained program structures. Post-training, formulas and Python reach nearly identical accuracy (64.23% vs. 64.00%), while SQL still lags behind at 55.60%, largely due to its limited ability to handle numerical computation required in table reasoning. Although the final scores of formulas and Python are close, formulas maintain a consistent edge. Beyond their strong performance, spreadsheet formulas are shorter, easier to read, and more beginner-friendly. These qualities make them not only effective but also practical and accessible as a symbolic tool for table reasoning tasks.

## 6 CONCLUSION

In this paper, we introduced *Formula Tuning* (Fortune), a reinforcement learning framework that trains language models to generate executable spreadsheet formulas for table understanding task. Our findings highlight the promise of formula-driven learning in enhancing reasoning capabilities of language models on tabular tasks. Limitations and future work are discussed in Appendix B.

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

# Contents of Appendix

## A  ETHICS STATEMENT

Formula Tuning (Fortune) introduces a reinforcement learning framework that enhances symbolic reasoning for table understanding via spreadsheet formulas. By improving the ability of language models to reason over tabular data with verifiable, executable outputs, our work offers substantial benefits in domains where transparency and precision are essential—such as education, scientific analysis, finance, and public policy. Executable formulas can provide interpretable and auditable reasoning steps, potentially increasing user trust and reliability in AI-generated decisions involving structured data.

However, these capabilities also introduce potential risks. If applied carelessly, formula generation may amplify biases present in training data or propagate subtle numerical errors. Moreover, spreadsheet formulas are deeply embedded in productivity workflows, and inaccurate generation at scale could lead to downstream harms (e.g., miscalculated budgets or flawed data reports). Furthermore, since symbolic reasoning via formulas may be more accessible in high-resource languages or domains with well-structured spreadsheets, deployment in low-resource settings could exacerbate inequalities in model performance and accessibility.

To mitigate such risks, we recommend several safeguards for future use of Fortune and similar symbolic reasoning systems. First, generated formulas should undergo verification through deterministic execution engines to ensure correctness. Second, evaluations should be conducted across diverse domains and spreadsheet structures, particularly including noisy or adversarial formats. Third, human-in-the-loop validation should be used in high-stakes applications (e.g., healthcare or financial audits) to ensure interpretability and safety. Finally, we advocate for transparent reporting of formula generation limitations and the inclusion of provenance indicators that show how a particular output was derived, enabling error tracing and accountability.

## B  LIMITATIONS AND FUTURE WORK

While Fortune demonstrates strong performance, several limitations remain and suggest promising directions for future research.

**Limited datasets and experimental coverage.** Due to the resource-intensive nature of reinforcement learning, our evaluation is limited to a few representative public datasets, which primarily consist of clean and well-structured tables. This may not fully capture real-world scenarios, where spreadsheets often contain noisy, irregular, or complex two-dimensional layouts. Additionally, we experimented with only a limited set of base models and reinforcement learning algorithms (e.g., PPO). Nonetheless, we believe the experiments conducted in this work sufficiently demonstrate the effectiveness of our approach. Future work should explore a broader range of model sizes, architectures, and reinforcement learning algorithms across different downstream scenarios, such as applying Formula Tuning to larger models to achieve even better performance.

**Applicability to broader table understanding tasks.** Our method assumes that answers can be fully derived from tabular data via executable formulas, which holds for many symbolic reasoning tasks. However, this assumption may not extend to tasks involving free-form text, multi-modal inputs, or ambiguous supervision. Nonetheless, formulas may still serve as useful intermediate representations, auxiliary objectives, or reasoning grounding mechanisms in such settings. Investigating how formula tuning can benefit or integrate with these broader tasks is an important direction.

**Extensions to other formula-related tasks.** Executable formulas are central not only to reasoning but also to related tasks such as formula completion, correction, and refilling. These tasks could benefit from multi-task learning or joint training alongside formula reasoning. Conversely, using these tasks as pre-training objectives may also enhance symbolic reasoning capabilities via formula. Exploring how these tasks can be unified within a single framework could lead to more powerful and general-purpose symbolic table models.

**Cold-start challenges in reinforcement learning.** For base models with limited symbolic reasoning capabilities and minimal knowledge of spreadsheet formulas, reinforcement learning from scratch can be unstable. In our experiments, we mitigated this by using the same training corpus for both SFT cold-start and RL. However, curating independent and high-quality cold-start corpora and identifying optimal initialization checkpoints for reinforcement learning remain open

challenges.Furthermore, reinforcement learning itself is inherently unstable. Developing practical techniques to stabilize training and improve performance remains a critical area for exploration. Enhancing warm-up strategies and training stability could lead to significantly better RL outcomes.

**Reward design for formula optimization.** Our current reward signal is based solely on binary execution accuracy. While simple and effective, it overlooks important factors such as formula efficiency, token redundancy, and partial credit. Future work can incorporate more fine-grained reward shaping, including length penalties or structure-aware scoring, to improve both learning stability and the quality of generated formulas.

These limitations point to several promising directions for future research: (1) scaling Formula Tuning to diverse domains and tasks, (2) exploring joint learning of symbolic tasks, (3) developing more stable and adaptive reinforcement learning strategies, and (4) advancing reward engineering for structured output generation.

## C  SUPPLEMENTARY PROOFS

### C.1  PROOF OF SYMBOLIC REASONING SUPERIORITY

**Lemma 1** (Reward Decomposition). *Let the reward be defined as $r(a \mid s) = \mathbb{1}[a = a^\star(s)]$, where $a^\star(s)$ denotes the ground-truth answer.*

*For textual reasoning, the expected reward is:*

$$\mathbb{E}_{a \sim \pi_\theta^{\mathrm{txt}}}[r(a \mid s)] = \sum_a \pi_\theta^{\mathrm{txt}}(a \mid s) \cdot \mathbb{1}[a = a^\star(s)], \tag{8}$$

*which represents the probability of generating both a logically valid reasoning path and a numerically correct final answer.*

*For symbolic reasoning, the model generates a formula $f$, which is executed to produce an answer $a = \mathrm{exec}(f, \mathbb{T})$. The expected reward becomes:*

$$\mathbb{E}_{f \sim \pi_\theta^{\mathrm{sym}}}[r(\mathrm{exec}(f, \mathbb{T}) \mid s)] = \sum_f \pi_\theta^{\mathrm{sym}}(f \mid s) \cdot \mathbb{1}[\mathrm{exec}(f, \mathbb{T}) = a^\star(s)]. \tag{9}$$

*This corresponds to the probability of generating a valid reasoning path and a formula that yields the correct answer. Importantly, any formula that produces the correct output receives full reward, regardless of whether it matches the canonical ground-truth formula.*

**Assumption 1** (Symbolic Reasoning Setting)**.**

1. The formula executor is sound and complete with respect to the formula language $\mathcal{F}$.

2. All symbolic outputs are executed deterministically and without numerical error.

3. Both textual and symbolic policies are assumed capable of representing valid high-level solution strategies in their respective formats, namely text or formula.

*Proof.* Let $E_1$ denote the event that the model selects a correct high-level reasoning plan—i.e., a valid logical strategy that, if accurately followed, can lead to the correct answer.

By Assumption 1 (3), both the symbolic policy $\pi_\theta^{\mathrm{sym}}$ and the textual policy $\pi_\theta^{\mathrm{txt}}$ are assumed capable of producing such high-level plans:

$$\mathrm{P}_{\mathrm{sym}}[E_1] = \mathrm{P}_{\mathrm{txt}}[E_1]. \tag{10}$$

We now compare how these two policies execute the same plan downstream.

- **Symbolic reasoning.** After selecting a correct high-level plan, the symbolic policy proceeds by emitting a formal expression—typically a spreadsheet formula $f$—that directly encodes the solution. This formula is then passed to an external executor, which deterministically computes the final answer $a = \mathrm{exec}(f, \mathbb{T})$. Under Assumptions 1 (1) and (2), if the plan is correct, the execution will reliably yield the correct answer $a^\star(s)$. Thus, the expected reward under the symbolic policy is:

$$\mathbb{E}_{a \sim \pi_\theta^{\mathrm{sym}}}[r(a \mid s)] = \mathrm{P}[E_1]. \tag{11}$$

- **Textual reasoning.** In contrast, after selecting the same correct high-level plan, the textual policy must verbalize the intermediate reasoning steps and compute results step-by-step in free text. This includes performing arithmetic, maintaining numerical precision, and formatting the final answer string. Let $E_2$ denote the event that all intermediate computations and the final output are accurate. Then, the expected reward under the textual policy is:

$$\mathbb{E}_{a \sim \pi_\theta^{\text{txt}}}[r(a \mid s)] = \text{P}[E_1] \cdot \text{P}[E_2 \mid E_1]. \tag{12}$$

Unlike symbolic execution, this textual process is inherently fragile. Errors in numerical calculations, token prediction, or formatting can easily lead to incorrect final answers, resulting in a reward of 0.

Since $\text{P}[E_2 \mid E_1] \leq 1$, we conclude:

$$\mathbb{E}_{a \sim \pi_\theta^{\text{txt}}}[r(a \mid s)] \leq \text{P}[E_1] = \mathbb{E}_{a \sim \pi_\theta^{\text{sym}}}[r(a \mid s)], \tag{13}$$

which completes the proof. $\qquad\square$

### C.2 PROOF OF RL SUPERIORITY

**Assumption 2** (SFT Setting).

1. **Sufficient expressivity.** The model class $\{\pi_\theta(\cdot \mid s)\}$ is expressive enough to represent the teacher policy $\pi_g(\cdot \mid s)$ (a stronger model, e.g., GPT-4o), in the sense that

$$\inf_\theta \mathbb{E}_{s \sim p(s)} [D_{\text{KL}} (\pi_g(\cdot \mid s) \parallel \pi_\theta(\cdot \mid s))] = 0. \tag{14}$$

2. **Global optimization.** The optimization algorithm converges to a global optimum of the supervised fine-tuning (SFT) objective.

3. **Data sufficiency.** As the number of training examples $N \to \infty$, the empirical distribution $\hat{p}(s, a)$ converges almost surely to the true data-generating distribution $p(s) \pi_g(a \mid s)$.

**Lemma 2** (MLE Minimizes KL Divergence). *Maximum likelihood estimation (MLE) corresponds to minimizing the Kullback–Leibler (KL) divergence between the teacher policy $\pi_g$ and the model policy $\pi_\theta$. For any fixed input $s$, we have:*

$$\mathbb{E}_{a \sim \pi_g} [- \log \pi_\theta(a \mid s)] = H (\pi_g(\cdot \mid s)) + D_{\text{KL}} (\pi_g \parallel \pi_\theta), \tag{15}$$

*where $H(\pi_g)$ denotes the entropy of the teacher policy. Thus, maximizing the log-likelihood of $\pi_\theta$ under samples from $\pi_g$ is equivalent to minimizing the KL divergence from $\pi_g$ to $\pi_\theta$.*

The proof of Lemma 2 is provided in Appendix C.3.

**Lemma 3** (Convergence of SFT and Reward Upper Bound). *Let $s = (\mathbb{T}, q) \in \mathcal{S}$, where $\mathbb{T}$ is the input table and $q$ is the natural language question. Suppose the model generates a formula $f \sim \pi_\theta(\cdot \mid s)$, and let the final answer be computed deterministically as $a = \text{exec}(f, \mathbb{T})$.*

*Under Assumption 2, the optimal supervised fine-tuning (SFT) policy*

$$\pi_{\theta^\star} = \arg \max_\theta \mathbb{E}_{s \sim p, f \sim \pi_g} [\log \pi_\theta(f \mid s)] \tag{16}$$

*satisfies*

$$\pi_{\theta^\star}(f \mid s) = \pi_g(f \mid s) \quad \text{for almost every } s \in \mathcal{S}. \tag{17}$$

*Consequently, for the reward function $r(a \mid s) = \mathbb{1}[a = a^\star(s)]$, we have:*

$$\mathbb{E}_{s \sim p, f \sim \pi_{\theta^\star}} [r(\text{exec}(f, \mathbb{T}) \mid s)] \leq \mathbb{E}_{s \sim p, f \sim \pi_g} [r(\text{exec}(f, \mathbb{T}) \mid s)]. \tag{18}$$

The proof of Lemma 3 is provided in Appendix C.4.

*Remark* 3 (SFT Bound). This result shows that supervised fine-tuning (SFT), even under ideal assumptions of expressivity, optimization, and data sufficiency, can at most replicate the performance of the teacher policy. It thus establishes a theoretical upper bound on the expected task reward achievable by SFT alone.

**Assumption 3** (RL Exploration). For each input $s \in \mathcal{S}$, we assume that the policy distribution $\pi_\theta(\cdot \mid s)$ assigns non-zero probability mass to at least one correct action with reward $r(a^\star \mid s) = 1$. This does not require the policy to sample a correct action at every step, only that the support of the distribution includes some high-reward actions, so they may be discovered over the course of training.

*Proof.* Let $a^\star(s)$ be the ground-truth answer for input $s$, and suppose that the teacher policy $\pi_g(f \mid s)$ covers only a strict subset of all possible formulas $f$ such that $\mathrm{exec}(f, \mathbb{T}) = a^\star(s)$.

By Lemma 3, supervised fine-tuning under ideal assumptions can at best match the expected reward of $\pi_g$:

$$\mathbb{E}_{s \sim p, \, f \sim \pi_{\theta^\star}} \left[ r(\mathrm{exec}(f, \mathbb{T}) \mid s) \right] = \mathbb{E}_{s \sim p, \, f \sim \pi_g} \left[ r(\mathrm{exec}(f, \mathbb{T}) \mid s) \right]. \tag{19}$$

Now consider an RL policy $\pi_\theta^{\mathrm{RL}}$. Under Assumption 3, the RL policy explores the full action space and assigns non-zero probability to correct formulas $f'$ that are not in the support of $\pi_g$ but still satisfy $\mathrm{exec}(f', \mathbb{T}) = a^\star(s)$.

As the reward function $r(a \mid s)$ depends solely on execution correctness, and not formula structure, RL is able to collect reward on these additional correct actions that $\pi_g$ does not generate. Therefore,

$$\mathbb{E}_{s \sim p, \, f \sim \pi_\theta^{\mathrm{RL}}} \left[ r(\mathrm{exec}(f, \mathbb{T}) \mid s) \right] > \mathbb{E}_{s \sim p, \, f \sim \pi_g} \left[ r(\mathrm{exec}(f, \mathbb{T}) \mid s) \right], \tag{20}$$

which implies the desired result. $\qquad\square$

### C.3 PROOF OF MLE MINIMIZES KL DIVERGENCE

*Proof.* By definition of KL divergence and entropy:

$$\begin{aligned}
\mathbb{E}_{a \sim \pi_g} \left[ -\log \pi_\theta(a \mid s) \right] &= -\sum_a \pi_g(a \mid s) \log \pi_\theta(a \mid s) \\
&= -\sum_a \pi_g(a \mid s) \log \frac{\pi_\theta(a \mid s)}{\pi_g(a \mid s)} - \sum_a \pi_g(a \mid s) \log \pi_g(a \mid s) \\
&= D_{\mathrm{KL}}(\pi_g \| \pi_\theta) + H(\pi_g).
\end{aligned} \tag{21}$$

$\qquad\square$

### C.4 PROOF OF CONVERGENCE OF SFT AND REWARD UPPER BOUND

*Proof.* **(i) KL minimization.** By Lemma 2, maximizing the expected log-likelihood

$$\mathbb{E}_{s \sim p(s), \, f \sim \pi_g(\cdot \mid s)} \left[ \log \pi_\theta(f \mid s) \right] \tag{22}$$

is equivalent to minimizing the expected Kullback–Leibler (KL) divergence from the teacher policy:

$$\mathbb{E}_{s \sim p(s)} \left[ D_{\mathrm{KL}}(\pi_g(\cdot \mid s) \| \pi_\theta(\cdot \mid s)) \right]. \tag{23}$$

**(ii) Convergence.** Under Assumption 2(1), the model class $\{\pi_\theta\}$ is expressive enough such that there exists some $\theta^\star$ satisfying

$$\inf_\theta \mathbb{E}_{s \sim p(s)} \left[ D_{\mathrm{KL}}(\pi_g(\cdot \mid s) \| \pi_\theta(\cdot \mid s)) \right] = 0. \tag{24}$$

Assumption 2(2) ensures the optimization algorithm converges to this global optimum, and Assumption 2(3) guarantees that the empirical distribution $\hat{p}(s, f)$ converges to the true distribution $p(s)\pi_g(f \mid s)$ as the sample size $N \to \infty$.

Therefore, at convergence,

$$D_{\mathrm{KL}}(\pi_g \| \pi_{\theta^\star}) = 0 \quad \text{almost everywhere}, \tag{25}$$

which implies pointwise equivalence between the student and teacher policies:

$$\pi_{\theta^\star}(f \mid s) = \pi_g(f \mid s) \quad \text{for almost every } s \in \mathcal{S}. \tag{26}$$

**(iii) Reward upper bound.** Let $r(a \mid s) = \mathbb{1}[a = a^\star(s)]$ be the task reward, where $a = \operatorname{exec}(f, \mathbb{T})$ is the executed output. Since execution is deterministic and the student mimics the teacher exactly, we have:

$$\mathbb{E}_{s \sim p, f \sim \pi_{\theta^\star}} \left[ r(\operatorname{exec}(f, \mathbb{T}) \mid s) \right] = \mathbb{E}_{s \sim p, f \sim \pi_g} \left[ r(\operatorname{exec}(f, \mathbb{T}) \mid s) \right]. \tag{27}$$

Thus, supervised fine-tuning under ideal assumptions can at best match the teacher's reward performance. In particular, this expected reward serves as an upper bound for what SFT can achieve when trained only on demonstrations from $\pi_g$. $\qquad\square$

# D SUPPLEMENTARY DISCUSSION OF METHODOLOGY

## D.1 TEXTUAL VS. SYMBOLIC REASONING IN TABLE UNDERSTANDING

In addition to the formal analysis in Section 3.2, we highlight several conceptual advantages of symbolic reasoning for table understanding:

- **Execution-based computation.** Symbolic reasoning externalizes computation through deterministic execution, separating high-level logical planning from low-level arithmetic or formatting operations.

- **Compositionality and structure.** Spreadsheet formulas offer compositional and type-aware representations, providing stronger structural priors than unstructured text.

- **Verifiability and transparency.** Symbolic outputs are interpretable and verifiable: they can be inspected, tested, reused, or debugged—enabling traceable and auditable reasoning processes.

- **Discrete action space.** The symbolic action space is bounded and discrete, which facilitates more stable exploration and optimization during training.

- **Robustness to token-level variability.** Unlike textual reasoning, which is prone to errors from exposure bias or numerical drift, symbolic reasoning delegates exact computation to the executor, reducing dependency on fragile token generation.

## D.2 SFT VS. RL IN SYMBOLIC TABLE REASONING

We also expand upon the discussion in Section 3.3, comparing supervised fine-tuning (SFT) and reinforcement learning (RL) for symbolic reasoning:

- **SFT limitations.** SFT imitates teacher demonstrations at the token level and struggles to generalize beyond the training distribution. It penalizes semantically correct but structurally different formulas, constraining exploration.

- **Reward-aligned optimization.** RL optimizes directly for task-level correctness using execution-based rewards, allowing the model to discover diverse yet valid solution strategies.

- **Support for many-to-one mappings.** Since different formulas can yield the same correct answer, RL naturally accommodates this multiplicity, whereas SFT often fails to reward such diversity.

- **Flexible reward shaping.** RL allows for auxiliary reward terms—such as penalties on length, syntactic constraints, or correctness under verification—which are difficult to incorporate in SFT.

- **Improved generalization.** By optimizing for semantic correctness rather than mimicking surface-level token patterns, RL enables the model to generalize more effectively in both in-distribution (ID) and out-of-distribution (OOD) scenarios, including novel question types, unseen table schemas, and structurally diverse formulas.

## D.3 PRACTICAL CHALLENGES OF FORMULA TUNING

While reinforcement learning (RL) offers significant advantages for symbolic table reasoning, it also introduces several practical challenges, especially under the **assumptions** outlined in Section 3.2 and 3.3.

- **Exploration bottlenecks.** Assumption 3 assumes that the RL policy can eventually explore correct formulas. However, the space of possible formulas is extremely large, and valid, executable ones are rare—especially at the start of training. This makes it difficult for the model to receive useful reward signals, leading to slow or unstable learning.

- **Limited symbolic priors.** Unlike supervised fine-tuning (SFT), RL does not benefit from direct examples of correct formulas. If the model lacks prior knowledge of spreadsheet syntax or symbolic structures, it may struggle to generate meaningful outputs. This weak starting point often results in inefficient exploration and poor early performance.

- **RL training instability.** When training from scratch, the model often produces repetitive, invalid, or meaningless formulas in the early stages, receiving no reward. This can cause unstable training and hinder convergence. Empirically, initializing with a supervised or pretrained model leads to more stable training and faster reward learning.

- **Sparse and coarse reward signals.** Execution-based rewards typically only indicate whether the final answer is correct or not, without offering any feedback on partially correct or structurally promising outputs. This makes it harder for the model to learn from near misses. Designing more informative reward functions—such as those based on formula structure or partial execution—remains an important direction.

Overcoming these challenges is essential for scaling *Formula Tuning* to more complex symbolic tasks, broader domains, and higher-capacity models. Future work may explore techniques such as curriculum learning, hybrid supervision, symbolic inductive priors, or multi-objective optimization to improve training stability and exploration efficiency.

# E  DETAILED SETTINGS OF EXPERIMENTS

**Models.** Our experiments include both open-source and proprietary models. For open-source models, we use *Qwen2.5-Coder$_{7B}$* (Qwen2.5-Coder-7B-Instruct, Apache 2.0 License) (Hui et al., 2024) and *LLaMA-3.1$_{8B}$* (LLaMA-3.1-8B-Instruct, Meta Llama 3 Community Licence) (AI, 2024). For proprietary models, we evaluate OpenAI's[1] *GPT-4o* (gpt-4o-2024-11-20), *GPT-4o-mini* (gpt-4o-mini-2024-07-18), and *O1* (o1-2024-12-17) as baselines.

**Datasets.** As shown in Table 4, we conduct experiments on seven diverse table understanding benchmarks: WikiTQ (Pasupat & Liang, 2015), TabFact (Chen et al., 2020), FinQA (Chen et al., 2021b), HiTab (Cheng et al., 2021), MultiHiertt (Zhao et al., 2022), AIT-QA (Katsis et al., 2021), and TableBench (Wu et al., 2025). These datasets vary in domain coverage, table structures, and question complexity, collectively spanning the full spectrum of table understanding tasks. For MultiHiertt, which contains multiple tables, we concatenate them vertically to form a single spreadsheet-like table. For training, we combine the first five datasets into a unified training corpus and train the model jointly on this merged set. Each dataset is then evaluated individually. All original training and test splits are preserved, except for TabFact, from which we randomly sample 10,000 examples to prevent its abundance of relatively simple binary QA examples from dominating or skewing the training. Among these benchmarks, AIT-QA and TableBench are considered out-of-distribution (OOD) evaluation sets, while the remaining datasets are treated as in-distribution (ID). The characteristics of each dataset are summarized below:

- **WikiTQ** is a Wikipedia-based table QA dataset with relatively simple factoid questions over relational tables.

- **TabFact** also uses Wikipedia tables but frames the task as fact verification, where each claim is labeled as either *true* or *false*.

- **FinQA** focuses on financial-domain tables and requires symbolic reasoning over semi-structured input that includes both pretext and posttext as additional context.

- **HiTab** contains hierarchical tables derived from statistical reports. While its structure is more complex than relational tables, the content is relatively straightforward.

- **MultiHiertt** involves multi-table reasoning over hierarchical tables in the financial domain, demanding both structural and symbolic reasoning.

---

[1]https://openai.com/policies/row-terms-of-use/

- **AIT-QA** consists of hierarchical tables from the airline domain. Although structurally rich, its questions tend to be simpler.
- **TableBench** features complex questions over relational tables drawn from various domains. Many questions require multi-step symbolic reasoning, making it the most challenging benchmark in our evaluation.

Table 4: Overview of the training data and table benchmarks used in this study.

| Evaluation Type | Dataset | # Train Data | # Test Data | Table Type | Domain | License | Source |
|---|---|---|---|---|---|---|---|
| In-Distribution | WikiTQ (Pasupat & Liang, 2015) | 13,753 | 4,217 | Relational | Wikipedia | CC-BY-SA-4.0 | Link |
| | TabFact (Chen et al., 2020) | 10,000 | 2,024 | Relational | Wikipedia | CC-BY-4.0 | Link |
| | FinQA (Chen et al., 2021b) | 6,251 | 1,147 | Relational | Finance | MIT | Link |
| | HiTab (Cheng et al., 2021) | 7,399 | 1,583 | Hierarchical | Statistical Reports | C-UDA 1.0 | Link |
| | MultiHiertt (Zhao et al., 2022) | 7,795 | 1,038 | Multiple & Hierarchical | Finance | MIT | Link |
| Out-of-Distribution | AIT-QA (Katsis et al., 2021) | – | 515 | Hierarchical | Airline | CDLA-Sharing-1.0 | Link |
| | TableBench (Wu et al., 2025) | – | 883 | Relational | Cross Domain | CC0-1.0 | Link |

**Table Encoding.** We adopt a table encoding method similar to SpreadsheetEncoder (Dong et al., 2024), which converts a table into a linearized markdown-style format. Each cell is represented by its spreadsheet address and value, forming text sequences such as `A1,Year|A2,Profit`. This encoding preserves both structural and content information, enabling the model to better understand cell-level references.

**Output Format.** Following the structured reasoning paradigm, the model is required to produce outputs in a two-stage format:

$$y = \underbrace{\langle\text{think}\rangle\, t\, \langle/\text{think}\rangle}_{\textit{reasoning trajectory}} \underbrace{\langle\text{answer}\rangle\, \{\text{json}\}\, \langle/\text{answer}\rangle}_{\textit{final answer}},$$

where $t$ is a free-form natural language reasoning process (i.e., the thinking process), and the answer block contains a JSON object from which the final prediction is extracted. This design enables decoupling the reasoning trajectory from the answer payload and facilitates more structured reward computation.

To encourage adherence to this format, we introduce a lightweight *format reward*. If the output fails to follow the required structure (e.g., malformed tags or unparseable JSON), the model receives a penalty of $-2$. If the format is valid and the answer can be successfully parsed from the JSON object, a small positive reward of $+0.1$ is added to the answer-level reward. Therefore, the final reward is as:

$$r_{\text{final}}(a \mid s) = r_{\text{ans}}(a \mid s) + r_{\text{fmt}}(a) \tag{28}$$

This reward shaping helps stabilize training and guide the model toward producing reliably structured outputs.

**Baselines.** We compare our proposed framework against a broad range of strong baselines, including both prompting-based and fine-tuning-based methods. To ensure a fair comparison, we require all methods to output short, deterministic answers rather than open-ended free-form text. Following this criterion, we exclude TableLLM (Zhang et al., 2025), which relies on a critique model for answer evaluation and does not produce a directly verifiable answer string. Prompting-based methods currently dominate the TableQA landscape, with most relying on large closed-source models for performance. We compare Fortune and Fortune++ with several representative methods in this category: Binder (Cheng et al., 2023), Dater (Ye et al., 2023), API-Assisted (Cao & Liu, 2025), Chain-of-Table (Wang et al., 2024), ReAcTable (Zhang et al., 2023), Norm-DP (Liu et al., 2023), TIDE (Yang et al., 2025), E5 (Zhang et al., 2024c), and SS-CoT (Zhao et al., 2024b). We also include Table-Master (Cao & Liu, 2025), a recent recipe-based prompting framework built on GPT-4o-mini. For fine-tuning-based methods, we select models specifically trained for table question answering tasks. These include TAPEX-Large (Liu et al., 2022), OmniTab (Jiang et al., 2022), TableLlama (Zhang et al., 2024a), TableGPT2 (Su et al., 2024), and TabAF (Wang et al., 2025), a recent strong method that combines formula generation and hybrid self-consistency. Our framework is evaluated under the same settings to ensure consistency and comparability across methods.

**RL Training.** We use Proximal Policy Optimization (PPO) for reinforcement learning (RL). The maximum prompt length is set to 8192 tokens, and the maximum response length is 512 tokens.

The critic model is initialized with the same weights as the actor model. The actor is trained with a learning rate of 1e-6, while the critic uses a slightly higher learning rate of 1e-5 to enable faster value estimation. We set the KL divergence coefficient to 0.001 to balance exploration and policy stability. The generation temperature is set to 0.6 to encourage a mix of determinism and diversity in the generated reasoning chains and formula outputs. The PPO mini-batch size is 64. We evaluate performance every 20 steps and report the results based on the best performance achieved on each dataset.

**SFT Training.** The supervised fine-tuning (SFT) training corpus is distilled from *GPT-4o* by prompting it with ground-truth answers, eliciting chain-of-thought reasoning followed by a final answer. We adopt a rejection-based fine-tuning (RFT) strategy (Yuan et al., 2023), retaining only examples where the generated answer exactly matches the ground truth. For symbolic reasoning tasks, correctness is determined by executing the generated formula and verifying that the resulting answer matches the expected output. This approach ensures high-quality supervision for fine-tuning. All SFT models are trained for 4 epochs, and we report results based on the checkpoint with the highest exact match accuracy on the test set. We use a learning rate of 2e-5 and a batch size of 64.

**Evaluation Inference.** For all models, including both open-source and proprietary ones, we use a temperature of 0, top-$k$ of 50, and top-$p$ of 0.7 during inference. Setting the temperature to 0 encourages deterministic outputs and improves stability in single-pass predictions. For self-consistency decoding in Fortune++, we use a higher temperature of 0.6 to promote diversity across multiple samples, enabling the model to better explore reasoning variations and improve final answer voting.

**Evaluation Metrics.** Following prior work (Pasupat & Liang, 2015; Cheng et al., 2021), we primarily use exact match (EM) as the evaluation metric, applying numeric tolerance when comparing numerical values. Official evaluation scripts are used whenever available to ensure consistency. For TabFact, which is formulated as a binary classification task, we report standard classification accuracy. Since our training objective aligns with evaluation, we also use exact match (EM) for answer reward calculation.

**Software.** We implement Fortune using Python 3.11, with the VERL framework (Sheng et al., 2024) serving as the core architecture for reinforcement learning and other supervised fine-tuning with language models. Our implementation utilizes VLLM (v0.8.3) (Kwon et al., 2023) for efficient LLM inference and generation, PyTorch (v2.4.0) with CUDA 12.8 for deep learning operations, and Ray (Moritz et al., 2018) for distributed training and inference. FlashAttention-2 (Dao et al., 2022) is integrated to accelerate attention computation. For proprietary LLMs, we access OpenAI models via the Microsoft Azure platform[2]. For formula execution, we use the open-source spreadsheet engine `formulas`[3] (EUPL 1.1+ License), which supports a wide range of standard spreadsheet operators. A representative list of symbolic operators used in our table reasoning framework is provided in Appendix J.

**Hardware.** All experiments are conducted on machines equipped with either NVIDIA A100 80GB PCIe or NVIDIA H100 80GB PCIe GPUs, along with 1.0 TB of RAM. For reinforcement learning (RL) training of open-source models, we use 8 × NVIDIA H100 80GB PCIe GPUs by default. For supervised fine-tuning (SFT) of open-source models, we use 4 × NVIDIA A100 80GB PCIe GPUs by default.

# F  ABLATION STUDY AND UPPER-BOUND PERFORMANCE OF FORTUNE++

We conduct an ablation study of Fortune++ to investigate the complementary roles and effectiveness of textual and symbolic reasoning. Table 5 reveals several instructive patterns.

**Combining textual and symbolic reasoning yields the most robust performance.** The balanced sampling strategy used by Fortune++ (five textual and five formula candidates) consistently outperforms both the pure-text and pure-formula variants across all benchmarks. This confirms that textual and symbolic reasoning address complementary error modes. Disabling either modality leads to substantial accuracy degradation, with drops of up to 18 percentage points (e.g., –18 pp on FinQA for text-only, –13 pp on AIT-QA for formula-only).

---

[2] https://azure.microsoft.com/
[3] https://github.com/vinci1it2000/formulas

Table 5: Performance comparison of TabAF, Fortune, and Fortune++ variants. Values in the table indicate accuracy (%). '-' indicates results not reported in the related paper. Gray rows represent upper-bound performance. Numbers in parentheses with a downward arrow (↓) indicate the performance drop relative to the default Fortune++ configuration. Top results are highlighted in dark blue.

| Method | | WikiTQ | TabFact | FinQA | HiTab | MultiHiertt | AIT-QA | TableBench |
|---|---|---|---|---|---|---|---|---|
| TabAF (Wang et al., 2025) | Upper Bound | 80.13 | 94.02 | - | 82.07 | - | - | - |
| | 5 Text | 61.42 | 81.47 | - | 74.24 | - | - | - |
| | 5 Formula | 64.20 | 67.54 | - | 74.87 | - | - | - |
| | 5 Text + 5 Formula | 74.72 | 83.99 | - | 78.41 | - | - | - |
| Fortune | Upper Bound | 77.35 | 96.49 | 72.01 | 79.91 | 54.82 | 88.93 | 43.26 |
| | 10 Formula | 67.05 | 85.08 | 62.16 | 69.74 | 40.85 | 80.39 | 35.22 |
| Fortune++ | Upper Bound | 93.62 | 99.51 | 91.02 | 95.01 | 71.29 | 98.06 | 61.16 |
| | 5 Text | 64.52 (↓18.02) | 85.18 (↓9.88) | 63.64 (↓16.83) | 74.48 (↓12.76) | 36.42 (↓15.31) | 83.30 (↓9.90) | 28.31 (↓16.65) |
| | 5 Formula | 66.48 (↓16.06) | 82.41 (↓12.65) | 61.99 (↓18.48) | 68.54 (↓18.70) | 39.60 (↓12.13) | 79.42 (↓13.78) | 34.65 (↓10.31) |
| | 5 Text + 5 Formula | **82.54** | **95.06** | **80.47** | **87.24** | **51.73** | **93.20** | **44.96** |

**Text and formula reasoning each excel in different scenarios.** Textual reasoning performs better on simpler table QA tasks, such as TabFact and AIT-QA, where natural language understanding and logical inference dominate. In contrast, formula-based reasoning excels on arithmetic-heavy or structured computation tasks like FinQA and MultiHiertt, where symbolic execution is crucial for deriving the correct answer. This division reinforces the importance of integrating both modalities for general-purpose table understanding.

**Reinforcement learning enhances symbolic reasoning beyond supervised fine-tuning.** Compared with TabAF—which uses the same backbone but is trained only with SFT—Fortune's RL variant achieves substantially stronger formula-only performance (e.g., 82.41% vs. 67.54% on Tab-Fact with 5 Formula). This suggests that reinforcement learning encourages the model to explore more reliable and executable reasoning paths, ultimately improving symbolic program quality.

**Many correct answers are lost due to naive majority voting.** The *upper-bound* rows show that Fortune++ frequently generates correct answers that are not selected by simple majority vote. The discrepancy between upper-bound and actual performance reaches 19 pp on MultiHiertt and 17 pp on TableBench, indicating considerable headroom for improved candidate selection through confidence-based aggregation or smarter reranking mechanisms.

**Complex, low-resource benchmarks remain the greatest challenge.** The largest performance gaps appear on structurally complex and low-resource datasets such as MultiHiertt and TableBench. These results highlight the limitations of current voting and reasoning mechanisms and point to future directions including symbolic planner integration, adaptive sampling, and confidence-calibrated answer selection.

# G    IMPACT ANALYSIS OF THE THINKING PROCESS IN FORMULA TUNING

Table 6 and Figure 3 highlight the impact of incorporating an explicit thinking process before generating formulas.

Table 6: Performance comparison with and without reasoning under Zero-Shot and RL settings across various datasets. Values in the table indicate accuracy (%).

| Method | WikiTQ | TabFact | FinQA | HiTab | MultiHiertt | AIT-QA | TableBench | Overall |
|---|---|---|---|---|---|---|---|---|
| ***Zero Shot*** | | | | | | | | |
| w/o Reasoning | 42.14 | 57.76 | 33.13 | 38.35 | 15.99 | 58.45 | 28.88 | 39.24 |
| w/ Reasoning | 38.96 | 52.52 | 34.44 | 32.60 | 15.90 | 52.43 | 26.05 | 36.13 |
| ***Reinforcement Learning (RL)*** | | | | | | | | |
| w/o Reasoning | 63.39 | 80.39 | 61.99 | 67.09 | 38.15 | 78.84 | 33.41 | 60.47 |
| w/ Reasoning | 67.80 | 84.19 | 62.16 | 71.19 | 41.72 | 81.17 | 35.45 | 63.38 |

**In the zero-shot setting, reasoning may hurts performance.** We observe that adding a reasoning trace before the formula generally leads to lower accuracy (e.g., 39.24% → 36.13% overall). This is likely because such *thought-first-then-formula* generation patterns are underrepresented in pretraining corpora. As a result, models tend to produce unnatural or error-prone reasoning steps, which negatively affect the final output. The limitations of chain-of-thought reasoning in certain scenarios have also been discussed in recent work (Sprague et al., 2025; Liu et al., 2024).

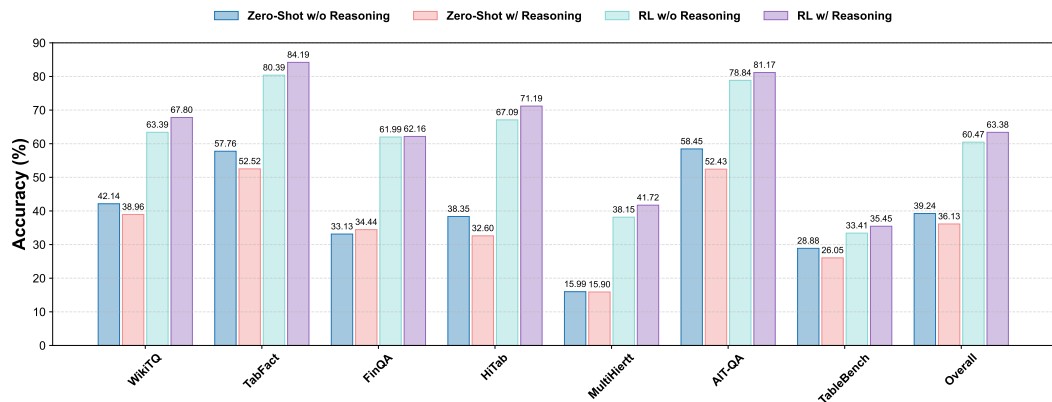

Figure 3: Performance comparison with and without explicit reasoning process under Zero-Shot and Reinforcement Learning (RL) settings across various datasets. Each group of bars shows the accuracy (%) achieved by four configurations: Zero-Shot without Reasoning, Zero-Shot with Reasoning, RL without Reasoning, and RL with Reasoning.

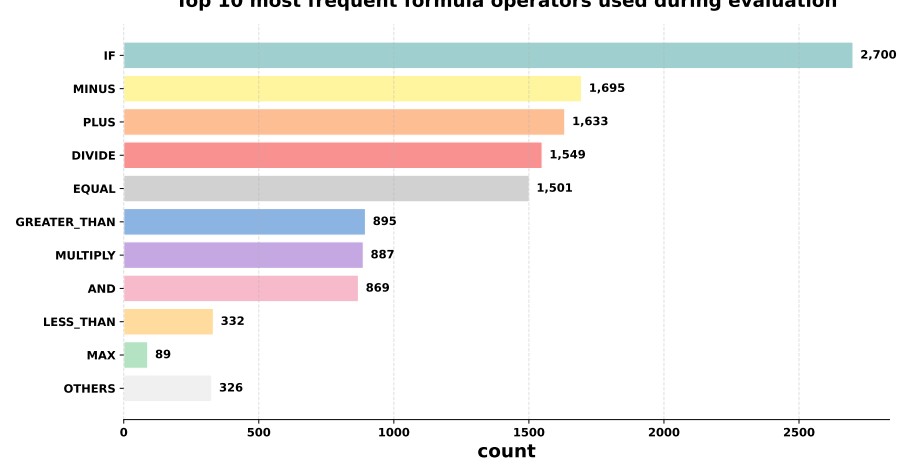

Figure 4: Top 10 most frequent formula operators used during evaluation.

**In the RL setting, reasoning significantly improves performance.** Once trained with our answer-based reward, the model begins to benefit from generating an explicit reasoning trace. The inclusion of a thinking process effectively expands the exploration space during policy optimization and encourages the model to break down complex table reasoning tasks into more manageable steps. This leads to consistent performance improvements across datasets (e.g., 60.47% → 63.38% overall), demonstrating that reasoning becomes a valuable asset—once the model has been properly trained to utilize it effectively.

# H STATISTICAL ANALYSIS OF GENERATED FORMULAS IN SYMBOLIC TABLE REASONING

We collect and analyze statistics of the formulas generated by our formula-tuned model during evaluation to better understand their structural properties.

Table 7 presents a quantitative summary of both table layout characteristics and the structural properties of generated formulas across seven widely used table understanding datasets. The table is divided into two parts: the first group (*Width*, *Height*, and *Area Size*) reflects the average struc-

Table 7: Statistics of generated formulas across different table understanding datasets.

| Dataset | Table Layout | | | Generated Formula | | |
|---|---|---|---|---|---|---|
| | Width | Height | Area Size | Length | # Operators | # Variables |
| WikiTQ | 6.28 | 19.46 | 121.68 | 21.61 | 0.84 | 1.50 |
| TabFact | 6.28 | 14.04 | 87.42 | 44.85 | 3.11 | 2.85 |
| FinQA | 3.92 | 18.09 | 70.80 | 18.84 | 1.98 | 2.94 |
| HiTab | 6.16 | 21.32 | 171.40 | 12.20 | 0.49 | 1.34 |
| MultiHiertt | 7.25 | 46.59 | 339.30 | 22.64 | 2.10 | 3.21 |
| AIT-QA | 5.62 | 13.86 | 81.91 | 5.67 | 0.11 | 1.89 |
| TableBench | 6.71 | 16.26 | 108.24 | 26.87 | 1.52 | 2.78 |

tural complexity of the input tables, while the second group (*Length*, *# Operators*, and *# Variables*) captures the syntactic and symbolic complexity of the generated formulas.

We observe substantial variation in table layout complexity. For example, MultiHiertt has by far the largest average table area (339.30), indicating its multiple and hierarchical format. In contrast, datasets like AIT-QA and FinQA involve relatively smaller or simpler tables, which may place less structural burden on the reasoning process. Notably, HiTab also exhibits a high area size, despite having fewer variables and a short formula length, suggesting that its challenge lies more in table structure than in formula richness.

In terms of generated formulas, TabFact stands out with the longest average formula length (44.85 characters) and the highest number of operators (3.11), indicating that its fact verification tasks typically require complex symbolic conditions. On the other hand, AIT-QA exhibits the shortest formulas with minimal operator usage (5.67 length, 0.11 operators), reflecting the dataset's relatively simple question types. Datasets like FinQA and MultiHiertt show high variable counts (around 3 per formula), which aligns with their multi-step reasoning nature involving multiple cell references. TableBench poses a greater challenge due to its combination of complex question intent and compositional reasoning demands. Although its average table size is moderate, the questions often require multi-step symbolic operations such as nested aggregations, comparisons, or indirect references—making it a strong testbed for evaluating deep reasoning ability.

These statistics provide important insights for model evaluation and reward design. First, different datasets pose very different reasoning demands—relying solely on benchmarks like WikiTQ or AIT-QA may underestimate a model's true symbolic capacity. Second, symbolic complexity (e.g., operator density) varies nontrivially across tasks, and therefore reward shaping mechanisms should adapt accordingly to avoid penalizing semantically necessary long formulas. Lastly, the disconnect between table area and formula length in datasets like HiTab implies that structural layout, rather than size alone, can be the main source of reasoning difficulty—an insight that can guide future benchmark construction and curriculum learning design.

Figure 4 presents the distribution of the most frequently used formula operators during evaluation. The conditional operator IF appears overwhelmingly often, with a count of 2,700, indicating that conditional reasoning is central to many table reasoning tasks. Arithmetic operators such as MINUS (1,695), PLUS (1,633), and DIVIDE (1,549) are also widely used, reflecting the numerical nature of many questions. Logical comparison operators like EQUAL, GREATER_THAN, and LESS_THAN occur frequently as well, suggesting that relational reasoning is also a common requirement. Less frequently used operators such as MAX and those grouped into the OTHERS category play a smaller role. Overall, the operator distribution highlights the need for models to support both arithmetic and logical reasoning, with a strong emphasis on conditional operations.

# I Case Study

## I.1 Textual vs. Symbolic Reasoning

We present representative examples comparing textual and symbolic reasoning methods in table understanding tasks.

As shown in Table 8, the textual approach performs better in this particular case. This is a simple counting question, so textual reasoning can easily enumerate the relevant items and output the correct answer (4). In contrast, the symbolic reasoning attempts to solve the problem via a more complex formula. Although the reasoning process is logically correct and the intent aligns with expectations, the actual formula execution produces an incorrect result due to implementation details—specifically, the presence of the string *nan* in the table being misinterpreted. This highlights that in certain corner cases, symbolic reasoning may not have fully mastered tool usage or aligned formula execution. In comparison, textual reasoning can sometimes bypass such pitfalls and arrive at the correct answer more robustly.

On the other hand, Table 9 illustrates a case where symbolic reasoning proves more effective. This is a moderately difficult classic counting problem with a ground truth answer of 13. Here, textual reasoning fails, producing an incorrect count (22), suggesting that its performance deteriorates as task complexity increases. In contrast, symbolic reasoning correctly formulates a count-based expression and arrives at the correct answer. The accompanying reasoning trace (*Think*) also clearly articulates the logic behind the computation, demonstrating the advantage of symbolic approaches in precise numerical tasks.

Table 8: Case study comparing textual and symbolic reasoning. In this example, textual reasoning yields the correct answer.

<table>
<tr><th>Method</th><th>Case</th></tr>
<tr><td>-</td><td>Question: rapper saigon has had at least () albums released in the U.S.?
Table:
<table>
<tr><th></th><th>A</th><th>B</th><th>C</th><th>D</th><th>E</th><th>F</th></tr>
<tr><td>1</td><td>Title</td><td>Year</td><td>Peak chart positions US</td><td>Peak chart positions US R&B</td><td>Peak chart positions US Rap</td><td>Album</td></tr>
<tr><td>2</td><td>Say Yes</td><td>2001</td><td>—</td><td>—</td><td>—</td><td>nan</td></tr>
<tr><td>3</td><td>Do You Know</td><td>2002</td><td>—</td><td>—</td><td>—</td><td>nan</td></tr>
<tr><td>4</td><td>Favorite Things</td><td>2004</td><td>—</td><td>—</td><td>—</td><td>nan</td></tr>
<tr><td>5</td><td>Pain In My Life (featuring Trey Songz)</td><td>2006</td><td>—</td><td>—</td><td>—</td><td>nan</td></tr>
<tr><td>6</td><td>C'mon Baby (featuring Swizz Beatz)</td><td>2007</td><td>—</td><td>—</td><td>—</td><td>nan</td></tr>
<tr><td>7</td><td>Gotta Believe It (featuring Just Blaze)</td><td>2009</td><td>—</td><td>—</td><td>—</td><td>Warning Shots 2</td></tr>
<tr><td>8</td><td>Bring Me Down</td><td>2010</td><td>—</td><td>—</td><td>—</td><td>nan</td></tr>
<tr><td>9</td><td>The Greatest Story Never Told</td><td>2011</td><td>—</td><td>—</td><td>—</td><td>The Greatest Story Never Told</td></tr>
<tr><td>10</td><td>Clap (featuring Faith Evans)</td><td>2011</td><td>—</td><td>—</td><td>—</td><td>The Greatest Story Never Told</td></tr>
<tr><td>11</td><td>Not Like Them (featuring Styles P)</td><td>2012</td><td>—</td><td>—</td><td>—</td><td>The Greatest Story Never Told Chapter 2: Bread and Circuses</td></tr>
<tr><td>12</td><td>Best Thing That I Found (featuring Lecrae and Corbett)</td><td>2012</td><td>—</td><td>—</td><td>—</td><td>The Greatest Story Never Told Chapter 2: Bread and Circuses</td></tr>
<tr><td>13</td><td>Best Mistake (featuring G. Martin)</td><td>2013</td><td>—</td><td>—</td><td>—</td><td>The Greatest Story Never Told Chapter 3: The Troubled Times of Brian Carenard</td></tr>
</table>
Groundtruth Answer: 4</td></tr>
<tr><td>Symbolic</td><td>Think: To compute how many unique albums Saigon has released, I examine column F (F2:F13), which lists album names. Some of these cells contain "nan", which I treat as missing values. I only count the unique non-missing values. A standard formula to count unique non-empty values is: =SUMPRODUCT(1/COUNTIF(F2:F13,F2:F13)) This formula works because COUNTIF returns the number of times each value appears; taking 1 over that and summing gives the count of distinct values. This will count duplicates only once and ignore empty cells or "nan" if treated as the same placeholder.
Generated Formula: =SUMPRODUCT(1/COUNTIF(F2:F13,F2:F13))

Execution Result: 5

Execution Accuracy: 0.0</td></tr>
<tr><td>Textual</td><td>Think: We are asked to find out how many albums rapper Saigon has released in the U.S. From the table, we look at the 'Album' column (column F), which lists the associated album for each song. We ignore entries that are 'nan' or empty, as those do not correspond to a known album. From F2 to F13, we extract the non-nan values: - Warning Shots 2 - The Greatest Story Never Told - The Greatest Story Never Told Chapter 2: Bread and Circuses - The Greatest Story Never Told Chapter 3: The Troubled Times of Brian Carenard These are 4 distinct album names, so the answer is 4.
Generated Answer: 4

Answer Accuracy: 1.0</td></tr>
</table>

Table 9: Case study comparing textual and symbolic reasoning. In this example, symbolic reasoning yields the correct answer.

| Method | Case |
|---|---|
| - | **Question:** *How many elections did the party win 75 or more seats?* 
 **Table:** 

 *(spreadsheet table as below)* 

 **Groundtruth Answer:** 13 |
| Textual | **Think:** *- A2, 1927 (sep): 57 seats - A3, 1932: 72 seats - ... - A26, 2002: 81 seats - A27, 2007: 77 seats - A28, 2011: 20 seats (less than 75) Counting the rows where the "seats" column is 75 or more: - 1927 (sep): 57 - 1932: 72 - ... - 2002: 81 - 2007: 77 Counting these, we find 22 elections where the party won 75 or more seats.* 
 **Generated Answer:** 22 

 **Execution Accuracy:** 0.0 |
| Symbolic | **Think:** *1. To answer this question, I need to look into the table provided in the question. The question is asking to determine how many times a party won 75 or more seats in various elections based on a given table. 2. Since the question is focused on the number of seats won, I need to look for a column in the table that contains the number of seats each party won in each election. 3. Given that the question is asking to count the number of elections where the party won 75 or more seats, I will need to look for a column that contains the seat numbers and then count how many of those seat numbers are 75 or greater. 4. Since the question is not providing a specific column in the table for seat numbers, I will need to infer that the seat numbers are contained in column D of the provided table, which is titled "seats".* 
 **Generated Formula:** `=COUNTIF(D2:D28, ">=75")` 

 **Execution Result:** 13 

 **Execution Accuracy:** 1.0 |

The table shown in the Question cell:

| | A | B | C | D | E |
|---|---|---|---|---|---|
| 1 | Election | Dáil | Share of Votes | Seats | Total Seats |
| 2 | 1927 (Jun) | 5th | 26.20% | 44 | 153 |
| 3 | 1927 (Sep) | 6th | 35.20% | 57 | 153 |
| 4 | 1932 | 7th | 44.50% | 72 | 153 |
| 5 | 1933 | 8th | 49.70% | 76 | 153 |
| 6 | 1937 | 9th | 45.20% | 68 | 138 |
| 7 | 1938 | 10th | 51.90% | 76 | 138 |
| 8 | 1943 | 11th | 41.80% | 66 | 138 |
| 9 | 1944 | 12th | 48.90% | 75 | 138 |
| 10 | 1948 | 13th | 41.90% | 67 | 147 |
| 11 | 1951 | 14th | 46.30% | 68 | 147 |
| 12 | 1954 | 15th | 43.40% | 65 | 147 |
| 13 | 1957 | 16th | 48.30% | 78 | 147 |
| 14 | 1961 | 17th | 43.80% | 70 | 144 |
| 15 | 1965 | 18th | 47.70% | 72 | 144 |
| 16 | 1969 | 19th | 44.60% | 74 | 144 |
| 17 | 1973 | 20th | 46.20% | 68 | 144 |
| 18 | 1977 | 21st | 50.60% | 84 | 148 |
| 19 | 1981 | 22nd | 45.30% | 77 | 166 |
| 20 | 1982 (Feb) | 23rd | 47.30% | 81 | 166 |
| 21 | 1982 (Nov) | 24th | 45.20% | 75 | 166 |
| 22 | 1987 | 25th | 44.20% | 81 | 166 |
| 23 | 1989 | 26th | 44.20% | 77 | 166 |
| 24 | 1992 | 27th | 39.10% | 68 | 166 |
| 25 | 1997 | 28th | 39.30% | 77 | 166 |
| 26 | 2002 | 29th | 41.50% | 81 | 166 |
| 27 | 2007 | 30th | 41.60% | 77 | 166 |
| 28 | 2011 | 31st | 17.40% | 20 | 166 |

## I.2 PERFORMANCE OF SFT VS. RL

We also compare models trained with supervised fine-tuning (SFT) and reinforcement learning (RL) in the context of symbolic table reasoning using formulas. As shown in Table 10, RL enables the language model to generate more accurate and executable formulas, leading to improved symbolic reasoning performance. In this example, the SFT model appears to merely memorize a reasoning pattern and produces a formula that is superficially related to the question but logically incorrect. Moreover, the intermediate reasoning process is highly interpretable and clearly aligned with the question's logic.

Table 10: Case study comparing performance of SFT vs. RL. In this example, RL yields the correct answer.

| Method | Case |
|---|---|
| - | **Question:** *What is the number of winning drivers that represented the U.S.?* 
 **Table:** 
  
 **Groundtruth Answer:** 15 |
| SFT | **Think:** *To determine the number of of winning States that represent, we need to count the occurrences of U.S. drivers in the range B1:B10. This will give us the total count of cells that contain the value "A1".* 
 **Generated Formula:** `=COUNTIF(B1:B10, A1)` 

 **Execution Result:** 0 

 **Execution Accuracy:** 0.0 |
| RL | **Think:** *1. To answer this question, I need to look into the table provided in the question. The question is asking to determine how many times a driver from the United States (U.S.) won a race based on a given table. 2. Since the question is focused on the winning drivers, I need to look for a column in the table that contains the names of the winning drivers. 3. Given that the question is asking to count how many times a U.S. driver won, I will need to look for a column that contains the names of the winning drivers and then count how many of those names are U.S. drivers. 4. Since the question is not providing a specific column or row in the table where the winning drivers' names are listed, I will need to assume that the winning drivers' names are listed in a column that I can reference in the table.* 
 **Generated Formula:** `=COUNTIF(G2:G18, "Rick Mears") + COUNTIF(G2:G18, "Al Unser, Jr.") + COUNTIF(G2:G18, "Michael Andretti") + COUNTIF(G2:G18, "Danny Sullivan")` 

 **Execution Result:** 15 

 **Execution Accuracy:** 1.0 |

## J  SPREADSHEET FORMULA OPERATORS IN SYMBOLIC TABLE REASONING

Spreadsheet formulas are Turing complete (Smalley, 2023), but in practice, only a core subset of functions is commonly used. To guide the language model toward effective formula generation and support future research, we identify a set of representative operators in Table 11. This table lists each operator's name, symbol, definition, and a representative example—primarily covering basic arithmetic and aggregation operations. Acknowledging the model's limited formula knowledge at the beginning of training, we explicitly introduce these operators during prompting, while still allowing the model to use any formula supported by our execution engine.

The formula operators can be viewed as the action space in reinforcement learning for symbolic table reasoning. The selected operators are designed to cover the majority of symbolic reasoning needs in table-based question answering, including row/column indexing, numerical aggregation, and conditional filtering. Focusing on a fixed set of operators facilitates interpretable error analysis and enables fine-grained tracking of formula usage patterns during both training and evaluation. This curated set also provides a natural foundation for curriculum learning strategies—starting with simpler operators and progressively introducing more complex ones, such as nested conditions and lookup functions.

Table 11: Representative spreadsheet formula operators in symbolic table reasoning: Symbols, Definitions, and Examples.

| Name | Symbol | Description | Example |
|------|--------|-------------|---------|
| PLUS | + | Adds two numbers together | =A1 + A2 |
| MINUS | − | Subtracts one number from another | =A1 − A2 |
| MULTIPLY | * | Multiplies two numbers together | =A1 * A2 |
| DIVIDE | / | Divides one number by another | =A1 / A2 |
| SUM | SUM | Sums a range of numbers | =SUM(A1:A10) |
| AVERAGE | AVERAGE | Calculates the average of a range of numbers | =AVERAGE(A1:A10) |
| COUNT | COUNT | Counts the number of numbers in a range | =COUNT(A1:A10) |
| MAX | MAX | Finds the maximum number in a range | =MAX(A1:A10) |
| MIN | MIN | Finds the minimum number in a range | =MIN(A1:A10) |
| EQUAL | = | Returns TRUE if the two values are equal | =A1 = A2 |
| NOT_EQUAL | <> | Returns TRUE if the two values are not equal | =A1 <> A2 |
| GREATER_THAN | > | Returns TRUE if the first value is greater than the second | =A1 > A2 |
| LESS_THAN | < | Returns TRUE if the first value is less than the second | =A1 < A2 |
| GREATER_THAN_OR_EQUAL | >= | Returns TRUE if the first value is greater than or equal to second | =A1 >= A2 |
| LESS_THAN_OR_EQUAL | <= | Returns TRUE if the first value is less than or equal to second | =A1 <= A2 |
| AND | AND | Returns TRUE if all arguments are TRUE | =AND(A1, A2) |
| OR | OR | Returns TRUE if any argument is TRUE | =OR(A1, A2) |
| NOT | NOT | Returns TRUE if the argument is FALSE | =NOT(A1) |
| IF | IF | Returns one value if a condition is TRUE and another if FALSE | =IF(A1 > 10, "Yes", "No") |
| TRUE | TRUE | Returns TRUE | =TRUE |
| FALSE | FALSE | Returns FALSE | =FALSE |
| INDEX | INDEX | Returns the value of a cell at a specific row and column | =INDEX(A1:A10, 1) |
| MATCH | MATCH | Returns the position of an item in an array (see syntax below) | =MATCH("value", A1:A10, 0) |

## K PROMPTS USED IN THE EXPERIMENTS

Figures 5, 6, 7, 8, and 9 illustrate the prompts used in our experiments across zero-shot inference, supervised fine-tuning (SFT), reinforcement learning (RL), and evaluation.

*Pre-text* and *Post-text* refer to optional unstructured context surrounding the table, such as the data description format used in FinQA (Chen et al., 2021b). *formula operator instruction* represents the textual representations and usage guidelines of the representative spreadsheet formula operators in symbolic table reasoning, as detailed in Appendix J.

These prompts serve as examples rather than optimal templates. They may vary across tasks and can be further optimized for better performance.

### Prompt for Symbolic Reasoning with Formula

*You are a helpful assistant.*

*# Task*
*You are an expert in writing Spreadsheet formulas given a table and a question.*
*You first think about the reasoning process in the mind and then provides the user with the answer.*
*Your task is to generate the correct spreadsheet formula to answer a given question, based on the provided table.*

*# Spreadsheet Formula Operator List*
*Below is a JSON list of commonly used formula operators, including their instructions and examples.*
*{formula_operator_instruction}*

*# Table*
*The table is represented as cell-value pairs, where each pair consists of a cell address and its content, separated by a comma (e.g., 'A1,Year').*
*Multiple cells are separated by a pipe symbol '|' (e.g., 'A1,Year|A2,Profit').*
*Empty cell of A1 can be represented as 'A1,|A2,Profit'.*

*Pre-text:*
*{pre_text}*

*Here is the table:*
*{table_content}*

*Post-text:*
*{post_text}*

*# Response Format*
*Show your reasoning within <think> </think> tags. Your final output must be in JSON format, enclosed in <answer> </answer> tags. For example:*
*<think>*
*[step-by-step reasoning]*
*</think>*
*<answer>*
*{{*
*"formula": "=......."*
*}}*
*</answer>*

*# Notes*
*1. For simple questions, if a direct cell reference is appropriate, simply return the formula as =CellAddress.*
*2. Construct the formula mainly using the provided operator symbols from the formula operator list.*
*3. You may either use cell references (cell addresses) in formulas or use the actual cell values directly.*
*4. Do not use the dollar sign ($) in addresses; use only formats like A1, A2, etc.*
*5. If a question has multiple answers, concatenate them using ", " as the separator. For example, use the formula `=A1 & ", " & A2 & ", " & A3` to produce a single string like `a, b, c`.*
*6. The execution result of the generated formula must be the direct final answer to the question.*

*Here is the question:*
*{question}*

*Let me write the spreadsheet formula with reasoning.*
*<think>*

Figure 5: Prompt for symbolic reasoning with formula. Blue text indicates placeholders for variables within the prompt.

## Prompt for Textual Reasoning

*You are a helpful assistant.*

*# Task*
*You are an expert in answering questions given a table.*
*You first think about the reasoning process in the mind and then provides the user with the answer.*
*Your task is to generate the correct answer to a given question, based on the provided table.*

*# Table*
*The table is represented as cell-value pairs, where each pair consists of a cell address and its content, separated by a comma (e.g., 'A1,Year').*
*Multiple cells are separated by a pipe symbol '|' (e.g., 'A1,Year|A2,Profit').*
*Empty cell of A1 can be represented as 'A1,|A2,Profit'.*

*Pre-text:*
*{pre_text}*

*Here is the table:*
*{table_content}*

*Post-text:*
*{post_text}*

*# Response Format*
*Show your reasoning within <think> </think> tags. Your final output must be in JSON format, enclosed in <answer> </answer> tags. For example:*
*<think>*
*[step-by-step reasoning]*
*</think>*
*<answer>*
*{{*
*"answer": "......"*
*}}*
*</answer>*

*# Notes*
*1. Use the values from the table in the reasoning process or answer the question.*
*2. If a question has multiple answers, concatenate them using ", " as the separator, e.g., "a, b, c".*
*3. Your answer cannot be the spreadsheet formula.*

*Here is the question:*
*{question}*

*Let me give the answer with reasoning.*
*<think>*

Figure 6: Prompt for textual reasoning. Blue text indicates placeholders for variables within the prompt.

## Prompt for Symbolic Reasoning with Python

*You are a helpful assistant.*

*# Task*
*You are an expert in writing Spreadsheet formulas given a table and a question.*
*You first think about the reasoning process in the mind and then provides the user with the answer.*
*Your task is to generate the correct Python code to answer a given question, based on the provided table.*

*# Table*
*The table is represented as cell-value pairs, where each pair consists of a cell address and its content, separated by a comma (e.g., 'A1,Year').*
*Multiple cells are separated by a pipe symbol '|' (e.g., 'A1,Year|A2,Profit').*
*Empty cell of A1 can be represented as 'A1,|A2,Profit'.*

*Pre-text:*
*{pre_text}*

*Here is the table:*
*{table_content}*

*Post-text:*
*{post_text}*

*# Response Format*
*Show your reasoning within <think> </think> tags. Your final output must be in JSON format, enclosed in <answer> </answer> tags. For example:*
*<think>*
*[step-by-step reasoning]*
*</think>*
*<answer>*
*{{*
*"code": "=......"*
*}}*
*</answer>*

*# Notes*
*1. Generate a Python code that can be executed to answer the question.*
*2. The result of executing the code should be the final answer.*
*3. You must output the Python code as a single line in the code field of the JSON, enclosed in triple backticks with the python tag (```python```).*
*4. If a question has multiple answers, concatenate them using ", " as the separator.*
*5. The input value for the table is already assigned to the variable 'table_data = [[...],[...],...]'.*
*6. The result of final answer must be assigned to a variable named 'answer'.*

*Here is the question:*
*{question}*

*Let me write the Python code with reasoning.*
*<think>*

Figure 7: Prompt for symbolic reasoning with Python. Blue text indicates placeholders for variables within the prompt.

## Prompt for Symbolic Reasoning with SQL

*You are a helpful assistant.*

*# Task*
*You are an expert in writing Spreadsheet formulas given a table and a question.*
*You first think about the reasoning process in the mind and then provides the user with the answer.*
*Your task is to generate the correct SQL query to answer a given question, based on the provided table.*

*# Table*
*The table is represented as cell-value pairs, where each pair consists of a cell address and its content, separated by a comma (e.g., 'A1,Year').*
*Multiple cells are separated by a pipe symbol '|' (e.g., 'A1,Year|A2,Profit').*
*Empty cell of A1 can be represented as 'A1,|A2,Profit'.*

*Pre-text:*
*{pre_text}*

*Here is the table:*
*{table_content}*

*Post-text:*
*{post_text}*

*# Response Format*
*Show your reasoning within <think> </think> tags. Your final output must be in JSON format, enclosed in <answer> </answer> tags. For example:*
*<think>*
*[step-by-step reasoning]*
*</think>*
*<answer>*
*{{*
*"sql": "=......"*
*}}*
*</answer>*

*# Notes*
*Generate a SQL query that can be executed onto the table to answer the question.*
*The result of executing the SQL query should be the final answer.*
*You must output the SQL query as a single line in the SQL field of the JSON.*
*The table name in the SQL query must be 'TMP_TABLE'.*

*Here is the question:*
*{question}*

*Let me write the SQL query with reasoning.*
*<think>*

Figure 8: Prompt for symbolic reasoning with SQL. Blue text indicates placeholders for variables within the prompt.

## Prompt for Symbolic Reasoning without Thinking Process

*You are a helpful assistant.*

*# Task*
*You are an expert in writing Spreadsheet formulas given a table and a question.*
*You need to provide the user with the answer directly.*
*Your task is to generate the correct spreadsheet formula to answer a given question, based on the provided table.*

*# Spreadsheet Formula Operator List*
*Below is a JSON list of commonly used formula operators, including their instructions and examples.*
*{formula_operator_instruction}*

*# Table*
*The table is represented as cell-value pairs, where each pair consists of a cell address and its content, separated by a comma (e.g., 'A1,Year').*
*Multiple cells are separated by a pipe symbol '|' (e.g., 'A1,Year|A2,Profit').*
*Empty cell of A1 can be represented as 'A1,|A2,Profit'.*

*Pre-text:*
*{pre_text}*

*Here is the table:*
*{table_content}*

*Post-text:*
*{post_text}*

*# Response Format*
*Your final output must be in JSON format, enclosed in <answer> </answer> tags. For example:*
*<answer>*
*{{*
*"formula": "=......"*
*}}*
*</answer>*

*# Notes*
*1. For simple questions, if a direct cell reference is appropriate, simply return the formula as =CellAddress.*
*2. Construct the formula mainly using the provided operator symbols from the formula operator list.*
*3. You may either use cell references (cell addresses) in formulas or use the actual cell values directly.*
*4. Do not use the dollar sign ($) in addresses; use only formats like A1, A2, etc.*
*5. If a question has multiple answers, concatenate them using ", " as the separator. For example, use the formula `=A1 & ", " & A2 & ", " & A3` to produce a single string like `a, b, c`.*
*6. The execution result of the generated formula must be the direct final answer to the question.*

*Here is the question:*
*{question}*

*Let me write the spreadsheet formula.*
*<answer>*

Figure 9: Prompt for symbolic reasoning with thinking process. Blue text indicates placeholders for variables within the prompt.

## L    NOTATION TABLE

Table 12 provides a comprehensive list of the notations used throughout this paper, along with their corresponding descriptions. This table serves as a quick reference to help readers better understand the concepts presented in our work.

Table 12: Notation used throughout the paper

| Notation | Description |
|---|---|
| *General* | |
| $s$ | Input instance, a pair $(\mathbb{T}, q)$ of table and question |
| $\mathbb{T}$ | Input table with $m$ rows and $n$ columns |
| $q$ | Natural-language query |
| $C_{i,j}$ | Cell at row $i$, column $j$ in the table |
| $m, n$ | Number of rows and columns in $\mathbb{T}$ |
| $a$ | Generated answer (textual or executed formula result) |
| $f$ | Spreadsheet formula generated by the model |
| $p(s)$ | Empirical distribution over table–query pairs |
| $r(a \mid s)$ | Reward function evaluating answer correctness |
| $a^\star(s)$ | Ground-truth answer for input $s$ |
| $\mathrm{exec}(f, \mathbb{T})$ | Deterministic executor applying $f$ to $\mathbb{T}$ |
| *Policies* | |
| $\pi_\theta(a \mid s)$ | LM generation policy parameterized by $\theta$ |
| $\pi_\theta^{\mathrm{txt}}$ | Textual reasoning policy (free-text answer) |
| $\pi_\theta^{\mathrm{sym}}$ | Symbolic reasoning policy (formula-based) |
| $\pi_g$ | Teacher policy in supervised fine-tuning |
| $\pi_{\theta^\star}^{\mathrm{SFT}}$ | Optimal SFT policy under Assumption |
| $\pi_\theta^{\mathrm{RL}}$ | Policy learned via reinforcement learning |
| *Objective and Metrics* | |
| $\mathbb{E}_{s \sim p,\, a \sim \pi}[\cdot]$ | Expectation under inputs and policy |
| $\mathbb{1}[\cdot]$ | Indicator function (1 if true, 0 otherwise) |
| *Assumptions and Events* | |
| $E_1$ | Event of selecting a correct high-level reasoning plan |
| $E_2$ | Event that all textual reasoning steps are correct |

## M    THE USE OF LARGE LANGUAGE MODELS (LLMS)

In this work, large language models (LLMs) were used *only to aid with writing and polishing the manuscript*. Specifically, LLMs were employed for grammar correction, phrasing suggestions, and improving readability. All research ideas, methodological contributions, theoretical analyses, and experiments were entirely conceived, designed, and executed by the authors without the involvement of LLMs. The authors take full responsibility for the scientific content of the paper.

