# OpenReview forum: "Fortune: Formula-Driven Reinforcement Learning for Symbolic Table Reasoning in Language Models"
_ICLR.cc/2026/Conference — ICLR 2026 Conference Withdrawn Submission_

### Official Review · Reviewer_btVc · 2025-10-22

**Soundness:** 2
**Presentation:** 4
**Contribution:** 3
**Rating:** 4
**Confidence:** 4

**Summary:**

The paper introduces **FORTUNE**, a reinforcement learning (RL) framework for training large language models (LLMs) to generate **spreadsheet formulas** as an explicit symbolic reasoning mechanism for table-based question answering. Instead of relying on annotated for-mulas, FORTUNE uses **binary answer correctness** as the reward signal, teaching the model to derive executable formulas that yield correct results when evaluated. The authors pro-vide **theoretical justifications** showing the superiority of symbolic over textual reasoning and RL over supervised fine-tuning (SFT), followed by **experiments** across seven table reasoning benchmarks. Results indicate substantial gains over both SFT and prior state-of-the-art (SOTA) systems, with the FORTUNE++ variant (combining symbolic and textual reason-ing) outperforming larger closed-source models such as OpenAI o1.

**Strengths:**

1. **Novel conceptual framing of formula-driven RL**
   The idea of representing symbolic table reasoning through spreadsheet formulas, rather than the more typical SQL or Python program synthesis, is both novel and practically motivated. The paper convincingly argues that formulas provide a more lightweight and accessible sym-bolic interface, with empirical results validating this claim.

2. **Comprehensive empirical evaluation**
   Results span seven datasets (WikiTQ, TabFact, HiTab, FinQA, MultiHiertt, AIT-QA, TableBench), covering both in-distribution and out-of-distribution settings. The comparisons are thorough—against prompting, supervised, and hybrid baselines—and the improvements are both consistent and meaningful.

3. **Clear analysis and ablation**
   The paper goes beyond headline results to provide analysis of textual vs symbolic reasoning, the effects of RL vs SFT, and the relative strengths of SQL, Python, and formula-based reason-ing. These comparative results lend strong credibility to the claims.

4. **Strong empirical impact**
   Achieving performance that surpasses OpenAI o1 with a 7B open model is a compelling re-sult and likely to attract interest from both academia and industry.

**Weaknesses:**

1. **Limited novelty in RL methodology**
   The reinforcement learning setup is a relatively standard application of PPO with scalar cor-rectness reward. While effective, the technical innovation on the RL side is incremental. The contribution is more conceptual (using formulas as the medium) than algorithmic.

2. **Limited theoretical depth**
   The two presented theorems (symbolic ≥ textual, RL ≥ SFT) are intuitive restatements of re-ward-maximization principles rather than new theoretical insights. Proofs lack quantitative bounds or convergence guarantees and do not address RL instability or reward-variance issues.

3. **Spreadsheet-formula limitations underexplored**
   While the paper highlights the flexibility of formulas, it underplays practical drawbacks—limited scalability on large tables and potential maintainability issues compared to structured systems like SQL/Pandas.

4. **Reward design and training stability**
   The reward structure (1 / 0.2 / 0) is coarse and heuristic, with no reported analysis of training stability, reward variance, or convergence behavior. It remains unclear how robust FORTUNE is to sparse or noisy rewards and whether different reward formulations would yield similar improvements.

5. **Experimental comparability**
   Some baselines (e.g., TabAF, TableGPT) may differ in data scale or compute budgets, mak-ing it unclear whether observed improvements stem purely from reinforcement learning or broader training differences. The paper reports its own configurations but cannot guarantee parity across external baselines.

**Questions:**

1. **Reward Sensitivity:**
   Have you experimented with alternative reward functions (e.g., graded rewards based on par-tial correctness or token-level formula similarity)? How sensitive is FORTUNE’s performance to the specific (1 / 0.2 / 0) reward scheme? Also, why did you choose PPO over newer RL variants such as GRPO or REINFORCE++?

2. **Training Stability:**
   Did you observe instability, mode collapse, or reward hacking during RL optimization? How did you ensure convergence, given the sparse binary reward?

3. **Fairness of Comparison:**
   Can you clarify whether FORTUNE and baseline models were trained on comparable data scales and compute resources? Were all models fine-tuned on the same merged corpus de-scribed in Section 4.1?

4. **Quantitative Effect of RL:**
   Beyond accuracy, do you track executability rate, average formula length, or entropy to show how RL qualitatively changes formula generation?

---

> ### Author Response · Authors · 2025-11-26
>
> Thank you for the reviewer’s constructive and insightful comments. We sincerely appreciate the feedback, which has helped us significantly strengthen the clarity and completeness of the paper. Below we address each concern in detail.
>
> ---
>
> ## **[W1] Limited novelty in RL methodology**
>
>
> Thank you for raising this point. We agree that our RL setup uses a standard PPO formulation with a scalar correctness reward, and we do not claim algorithmic novelty on the RL side. Our work is application-oriented and focuses on how RL can be effectively applied to symbolic table reasoning, rather than introducing a new RL algorithm.
>
> As you also noted, the core novelty of Fortune lies in its conceptual framing of formula-driven RL, rather than in the optimizer itself. Specifically:
> 1. Spreadsheet formulas as a symbolic action space: We introduce spreadsheet formulas as an explicit symbolic action space for RL, which is fundamentally different from the SQL/Python program-synthesis spaces used in prior work.
> 2. Lightweight, safe, and compositional symbolic interface: This design provides a structured but flexible symbolic medium that LLMs can efficiently explore, enabling reasoning behaviors that do not emerge under purely textual supervision.
> 3. Surprising empirical impact: We show that this symbolic action space allows a 7B open-source LM to surpass the performance of much larger proprietary models—demonstrating that the choice of symbolic medium can significantly amplify the effectiveness of RL in table reasoning.
>
> In short, while the RL algorithm itself is standard, the integration of RL with a spreadsheet-formula symbolic space is the primary contribution, and our empirical findings confirm that this conceptual design is both novel and highly impactful.
>
> ---
>
> ## **[W2] Robustness to Real-World Tables**
>
> We appreciate the reviewer’s feedback on the theoretical section. Our intent was not to introduce new RL theory, but to provide intuitive, principled explanations for why symbolic formulas provide more informative reward signals than textual reasoning, and why RL is preferable to SFT in this setting.
>
> We agree that the current statements are high-level. In the revision, we will expand our theoretical discussion by clarifying the assumptions underlying symbolic vs. textual reward informativeness, adding quantitative analysis to illustrate how executability constraints reduce the effective search space, and discussing reward variance and stability in symbolic action spaces with support from the new training curves we now include. We will also position our theorems as interpretive insights rather than formal convergence guarantees, consistent with prior reasoning-focused RL work.
>
> Our goal is to make clearer that the theory is intended to provide explanatory intuition rather than new RL foundations. The empirical evidence remains the primary support for the effectiveness of Fortune.
>
> ---
>
> ## **[W3] Spreadsheet-formula limitations underexplored**
>
> Thank you for raising this point. We agree that spreadsheet formulas have practical limitations compared to SQL/Pandas, particularly with respect to scalability on extremely large tables or multi-table relational operations. However, these limitations do not affect the setting of our work for several reasons.
>
> First, several of the benchmarks we evaluate—HiTab and AIT-QA in particular—already contain real-world tables with substantial size, hierarchical structures, and heterogeneous cell types. In these scenarios, we found that spreadsheet formulas are equally effective as SQL/Pandas-style operators for the types of single-table reasoning tasks these benchmarks require. Empirically, the executor handles all tables in our experiments efficiently, and we did not observe any scalability or maintainability issues.
>
> Second, Fortune is trained via reinforcement learning rather than large-scale SFT, meaning it does not depend on maintaining long formula pipelines or human-engineered symbolic programs. The model generates short, compositional formulas tailored to each query, which avoids maintainability concerns and naturally fits the complexity of the evaluated benchmarks.
>
> We agree that for very large relational databases or workloads involving multi-table joins, SQL/Pandas would be more appropriate. We will clarify this trade-off in the revision and emphasize that the spreadsheet-formula symbolic space is intentionally chosen to match the scope and requirements of current table QA benchmarks, where it performs on par with more heavyweight structured systems.
>
>
> ---

---

> ### Author Response · Authors · 2025-11-26
>
> ## **[W4 & Q1 & Q2] Reward Sensitivity and Training Stability**
>
> 1. Reward design.
> We experimented with several alternative reward designs. Our primary goal is to reward correct answers, so we tested partial-correctness rewards (e.g., cell-wise overlap) and token-level formula similarity. However, partial-credit signals consistently led to reward hacking, where the model learned to exploit intermediate structures without improving final correctness, resulting in worse downstream performance. Token-level similarity behaved similarly to SFT-style supervision and did not meaningfully guide symbolic reasoning.
> Moreover, using only two reward states—1 for correct formulas and 0 for everything else—created an extremely sparse reward signal, causing PPO to collapse into trivial textual outputs or short nonsensical formulas. To mitigate this, we introduce a small positive reward (0.2) for formulas that are executable but incorrect. This intermediate reward stabilizes early training: it encourages the model to first learn to produce syntactically valid formulas, thereby reducing the exploration space, accelerating convergence, and enabling a two-stage learning process (validity → correctness).
>
> 2. Sensitivity to the 1 / 0.2 / 0 scheme.
> We re-examined our training logs and observed similar trends when testing intermediate rewards of 0.1, 0.3, and 0.4. The final accuracy fluctuated by only ±0.03, but larger values caused reward hacking, while smaller values led to training instability. 0.2 consistently emerged as the most stable and effective setting across runs.
>
> 3. Choice of PPO.
> We use PPO simply because it is the most established and stable method for discrete, structured action spaces such as formula generation. Importantly, our contribution is RL-method–agnostic: the core insight of Fortune lies in demonstrating that LLMs can leverage formula-driven symbolic reasoning under reinforcement learning. Using a widely adopted baseline algorithm ensures that our improvements stem from the symbolic formulation rather than from a specific RL optimizer.
>
> 4. Training Stability
> We also note that, in contrast to many program-synthesis–style RL approaches that often suffer from instability due to sparse rewards or brittle action spaces, Fortune trains very stably in practice. The combination of a structured formula action space and our two-stage reward design leads to smooth optimization dynamics: we consistently observe stable executability rates, no mode collapse, and reliable convergence across runs. Training curves included in the revised version further illustrate that the RL process in Fortune is substantially more stable than what is typically reported in related symbolic reasoning settings.
>
> ---
>
> ## **[W5 & Q3] Fairness of Comparison**
>
>
> Thank you for the thoughtful question. We clarify the comparison setup as follows.
>
> First, all FORTUNE variants and our internal baselines (zero-shot, SFT, RL) are trained and evaluated under matched, or in some cases even lower, data scales and compute budgets, using a similar subset of publicly available datasets. This ensures that comparisons within our framework remain fully controlled and fair.
>
> For some external baselines, however, strict data and compute parity cannot be guaranteed. Several prior methods do not use the same dataset set, and some rely on additional large-scale pretraining or synthetic SFT corpora that we do not employ. For example, TableGPT2 leverages substantial domain-specific pretraining and large synthetic formula datasets far exceeding our data scale. Other baselines omit one or more datasets we include, or adopt heterogeneous fine-tuning pipelines. In certain cases, these methods use more data than our setup.
>
> Second, the primary methodological comparison in our work is not SFT vs. SFT across different papers, but rather RL vs. SFT vs. zero-shot under a controlled, shared base model. This is exactly what Table 1 presents: all three settings use the same 7B base model, the same datasets, and the same compute budget, providing a fair and isolated evaluation of FORTUNE’s contribution.
>
> Thus, although external baselines differ in their training configurations, the core findings of our work rely on consistent, controlled comparisons within the same model, demonstrating that the gains stem from the reinforcement learning framework rather than from differences in data or compute.
>
>
> ---

---

> ### Author Response · Authors · 2025-11-26
>
> ## **[Q4] Quantitative Effect of RL**
>
>
> We revisited the training logs to track several qualitative metrics during RL training to better understand how FORTUNE shapes formula-generation behavior. Using Qwen2.5-Coder-7B as a representative base model, we observe clear and consistent trends.
> 1. Executability rate increases substantially over training, rising from 0.353 → 1.00, indicating that the model first learns to produce syntactically valid formulas before optimizing for correctness.
> 2. Average formula length exhibits a non-monotonic pattern: it begins at 205 tokens, briefly decreases to 97 tokens, and eventually stabilizes around 272 tokens. We believe this reflects a natural progression in which the model initially learns shorter valid formulas and later composes longer, multi-step formulas to handle more complex reasoning.
> 3. Policy entropy decreases smoothly from 0.204 → 0.066, consistent with healthy PPO dynamics—high entropy facilitates early exploration, followed by gradual entropy reduction as the policy converges without collapse.
>
> These qualitative metrics support our claim that RL meaningfully reshapes the model’s behavior: it first stabilizes executability, then encourages structured symbolic composition, and ultimately converges to a low-entropy, high-validity formula-generation policy.
>
>
> ---
>
>
> Thank you again for your valuable feedback. We believe these updates strengthen the paper and adequately address the reviewer’s concerns.
>
> **Sincerely,**
> Authors of *Fortune*

---

### Official Review · Reviewer_LkyD · 2025-10-30

**Soundness:** 3
**Presentation:** 3
**Contribution:** 2
**Rating:** 4
**Confidence:** 2

**Summary:**

This paper propose a LLM framework that firstly use symbolic reasoning in reinforcement learning. By using the correctness of formula execution results as the reward signal, the framework reduces dependence on supervised formula annotations, guiding models to learn formula derivation through reasoning.

**Strengths:**

- This paper is the first to use symbolic reasoning in RL and gains huge enhancement.
- The tabular dataset used is sufficiently large in scale and the experimental setup is detailed, effective and complete enough.
- The proofs in appendix is detailed.

- This paper is the first to integrate symbolic reasoning (via spreadsheet formulas) with RL for table reasoning tasks, yielding significant performance improvements.
- The tabular datasets employed in this study are comprehensive, covering 7 diverse table reasoning benchmarks (e.g., WikiTQ, TabFact, FinQA). Moreover, the experimental setup is detailed and rigorous—including clear descriptions of model backbones, training protocols and evaluation metrics to ensure the reproducibility and validity of the results.
- The theoretical proofs presented in the appendix are comprehensive and detailed.

**Weaknesses:**

- Robustness is one of the key features in LLMs. However, this paper don't design experiments for this feature. When it comes to real world tables like NAN or nil data, it's important to know the results.
- While the paper reports performance under zero-shot, supervised fine-tuning, and RL settings, the evaluations are limited to unimodal LLMs. With the rapid advancement of multimodal large language models which can process tabular data alongside other modalities (e.g., image charts, textual descriptions in table captions)—the paper fails to extend its scope to MLLMs. This omission limits the framework’s generalizability to increasingly common multimodal table understanding tasks.

**Questions:**

- Symbolic methods are widely used in TableQA. For instance, GSM8K, a well-known mathematical reasoning dataset, can be converted into tabular formats to test table-based numerical reasoning. Given this, if the proposed Formula Tuning framework directly adopts formula-based parameterization instead of relying on LM-generated formula exploration, would this constitute a more effective modeling approach? If so, how might it impact the framework’s ability to handle complex multi-step reasoning tasks?
- For zero-shot reasoning, GPT-series models (e.g., GPT-4o, GPT-4o-mini) exhibit notably stronger performance compared to open-source models like Qwen2.5-Coder7B. First, could the authors explain the potential reasons for this significant zero-shot gap? Second, have the authors tested other state-of-the-art models such as Gemini in zero-shot settings to further validate whether the observed performance trends are specific to GPT-series models? Third, including more powerful models in the baseline comparisons would help better contextualize the superiority of the proposed FT framework. Are there plans to supplement such experiments?

---

> ### Author Response · Authors · 2025-11-26
>
> Thank you for the reviewer’s constructive and insightful comments. We sincerely appreciate the feedback, which has helped us significantly strengthen the clarity and completeness of the paper. Below we address each concern in detail.
>
> ---
>
> ## **[W1] Robustness to Real-World Tables**
>
> Thank you for highlighting the importance of robustness. Several of the datasets included in our evaluation—AIT-QA, HiTab, and MultiHiertt—are real-world table benchmarks that inherently contain hierarchical structures, missing values (NAN / nil), irregular formatting, and noisy cell contents. Fortune achieves strong performance on these datasets, which demonstrates that our method is naturally robust to such real-world imperfections.
>
> ---
>
> ## **[W2] Limitation to Unimodal LLMs**
>
> We acknowledge this limitation. Our work focuses specifically on symbolic table reasoning, where formulas operate strictly over structured textual table cells. Multimodal signals such as table images, charts, or captions do not directly support spreadsheet-style symbolic operations, and thus fall outside the scope of the current study.
>
> Importantly, Fortune is model-agnostic and does not depend on architectural assumptions; it only requires a model capable of generating formulas. Extending Fortune to multimodal LLMs—which must additionally handle OCR, layout parsing, and visual grounding—represents a meaningful and orthogonal research direction. We will mention this explicitly as promising future work.
>
> ---
>
> ## **[Q1] Formula-based parameterization vs. LM-driven formula exploration**
>
> Thank you for the thoughtful question. We appreciate the reviewer’s insight regarding symbolic methods in TableQA.
>
> In our setting, several of the benchmarks we evaluate—especially TableBench and FinQA—already involve highly complex, multi-step table reasoning. Importantly, our SFT baseline is trained on high-quality formula supervision generated by GPT-4 and filtered for correctness, which already represents a strong form of formula-based parameterization. Even under this favorable setup, we observe that cold-start RL-based formula exploration consistently outperforms SFT, with particularly large gains on harder questions. This indicates that allowing the model to freely explore and compose formulas is more effective than relying solely on predefined or supervised formula templates.
>
> Formula-based parameterization requires specifying or supervising fixed structural templates for formulas. While such templates can help constrain the search space, they are inherently limited: they cannot fully represent the diverse reasoning patterns across these datasets, nor support the multi-step, compositional reasoning paths that RL learns to discover. In contrast, LM-driven exploration enables the model to search over a flexible symbolic space and adaptively construct formulas conditioned on table layout and question semantics.
>
> In summary, although formula parameterization can be suitable for narrow or well-structured tasks, our empirical results—despite using strong GPT-4-generated supervision—show that exploration-based formula generation is substantially better suited for complex, multi-step table reasoning, which motivates the RL design in Fortune.
>
> ---
>
> ## **[Q2] Zero-shot Gap & More Baselines**
>
> 1. Why do GPT-series models achieve stronger zero-shot performance?
>
> We agree with the reviewer that GPT-4o and GPT-4o-mini exhibit significantly stronger zero-shot table reasoning than current open-source 7B models such as Qwen2.5-Coder-7B. This gap is expected: GPT models are trained on much larger and more diverse corpora, which likely include substantially richer exposure to formula-like patterns, mathematical reasoning traces, and structured tables. Combined with extensive instruction tuning, this provides stronger general reasoning priors that naturally translate into better zero-shot performance on table QA tasks. In contrast, open-source 7B models generally have more limited pre-training coverage and weaker mathematical/structured reasoning supervision.
>
> 2. Have the authors tested other models such as Gemini? Plans to supplement more baselines?
>
>
> We appreciate the suggestion. We have already evaluated all proprietary models that we currently have access to, including those available via our OpenAI API. Unfortunately, we do not have access to additional closed-source models such as Gemini, and therefore cannot conduct further evaluations at this time. We will explicitly mention this limitation in the revision and include broader evaluation on future foundation models as an important direction for subsequent work.
>
> ---
>
>
> Thank you again for your valuable feedback. We believe these updates strengthen the paper and adequately address the reviewer’s concerns.
>
> **Sincerely,**
> Authors of *Fortune*

---

### Official Review · Reviewer_9DMv · 2025-11-01

**Soundness:** 2
**Presentation:** 2
**Contribution:** 2
**Rating:** 4
**Confidence:** 4

**Summary:**

This paper introduces a training framework called Fortune to enhance reinforcement learning prompt models' ability to process tabular data. By having the model output executable formula outputs and using executability and correctness as reward functions, the model's tabular data processing capabilities significantly improve after reinforcement learning.

**Strengths:**

The proposed training method enables a 7B model to achieve strong tabular data processing capabilities, outperforming commercial models on certain datasets.

**Weaknesses:**

Support for tabular operations is insufficient; features like sorting should be added to better align with practical software like Excel. The specific implementation of the Formula Executor lacks description. Exploration of OOD (Operations Over Data) symbols is missing.

**Questions:**

1, Is there a specific reason for setting executable but incorrect results to 0.2 in the reward function?
2, A description of the Formula Executor's concrete implementation should be included.
3, It is recommended to test the model's performance before and after encountering data requiring operations not included in training to demonstrate its generalization ability.

---

> ### Author Response · Authors · 2025-11-26
>
> Thank you for the reviewer’s constructive and insightful comments. We sincerely appreciate the feedback, which has helped us significantly strengthen the clarity and completeness of the paper. Below we address each concern in detail.
>
> ---
>
> ## **[W1] Support for more advanced tabular operations (e.g., sorting)**
>
> Thank you for the insightful comment. Our current operator set is selected based on the taxonomy summarized in J. Spreadsheet Formula Operators in Symbolic Table Reasoning, focusing on the operators that are most frequently useful across the seven benchmarks we evaluate.
>
> In our experiments and case studies, we found that advanced operators such as SORT or RANK rarely improve performance on these datasets—mainly because the questions seldom require global reordering and are instead dominated by lookup, filtering, conditional aggregation, and arithmetic patterns. We also observed that large models almost never select these operators when given access to a broader operator pool, suggesting they are not essential for the types of reasoning present in the current benchmarks.
>
> That said, we fully agree that sorting and other higher-level operations are important for more complex real-world spreadsheets. We will incorporate a discussion of this design choice in the revision and include these advanced operators as part of future work for scenarios involving richer multi-step reasoning and larger relational tables.
>
> ---
>
> ## **[W2 & Q2] Description of the Formula Executor**
>
> Thank you for pointing this out. We apologize that the description in the main paper was too brief. As noted in Appendix E: Detailed Settings of Experiments, our implementation uses the open-source spreadsheet engine formulas (https://github.com/vinci1it2000/formulas), which provides a reliable and fully executable environment for spreadsheet-style formulas.
>
> To make this clearer, we will add a dedicated subsection in the revised version describing the concrete execution pipeline:
> - Parsing & AST Construction: the predicted formula is parsed into an abstract syntax tree using the formulas engine.
> - Reference Resolution: cell references (e.g., A1, A1:B5) are expanded into concrete index ranges and mapped to table cells.
> - Operator Support: the executor supports the restricted, pre-defined operator set used in our framework, including arithmetic, logical, conditional, lookup operators, etc.
> - Error Handling: execution errors (e.g., division-by-zero, invalid reference, type mismatch) are caught and normalized to a special #ERROR state, which yields reward 0.
> - Full Executability: all formulas are executed inside a sandboxed evaluator with no external side effects.
>
> ---
>
> ## **[Q1] Support for more advanced tabular operations (e.g., sorting)**
>
> Thank you for the question. In our preliminary experiments, we observed that using only two reward states—1 for correct formulas and 0 for all incorrect outputs—creates an extremely sparse reward signal, which causes PPO to collapse into producing short textual answers or trivial formulas. To mitigate this, we introduce a small positive reward (0.2) for formulas that are executable but yield incorrect results.
>
> This intermediate reward stabilize early training: it encourages the model to first learn to generate syntactically valid formulas, which dramatically reduces the exploration space and accelerates convergence. It can instruct the model first learns how to produce valid formulas, and then learns which formulas lead to correct answers. This staged progression leads to significantly better overall performance.
>
>
> ---
>
>
> ## **[Q3] OOD symbolic-operator generalization**
>
>
> Thank you for the insightful suggestion. In our current setup, the Formula Executor intentionally exposes only a pre-defined and restricted operator set. This controlled symbolic space is by design: it matches the operations actually required by all seven benchmarks, and consequently operator-level OOD cases do not naturally occur in these datasets.
>
> More importantly, Fortune is trained via reinforcement learning, where the model learns to actively explore and compose formulas within the provided operator set. We do not specify which operator should be used; the model must discover appropriate compositions on its own. Even within this restricted space, Fortune achieves strong OOD generalization across tables, domains, and question types. As shown in our experiments, RL significantly outperforms SFT on OOD splits, indicating that Fortune is not memorizing operator templates but is learning genuine compositional symbolic reasoning.
>
> We agree that incorporating operator-level OOD evaluations would be valuable for future work, especially for more complex real-world spreadsheets requiring richer operations.
>
> ---
>
>
> Thank you again for your valuable feedback. We believe these updates strengthen the paper and adequately address the reviewer’s concerns.
>
> **Sincerely,**
> Authors of *Fortune*

---

### Note · Authors · 2025-12-19

**Comment:**

We thank the reviewers for their valuable feedback and thoughtful comments. We will carefully incorporate the suggestions and revise the paper accordingly for a future submission. Therefore, we respectfully withdraw this submission.

**Withdrawal Confirmation:**

I have read and agree with the venue's withdrawal policy on behalf of myself and my co-authors.